# Structure of the TFIIIC subcomplex τA provides insights into RNA polymerase III pre-initiation complex formation

Matthias K. Vorländer[1,2], Anna Jungblut [1,2], Kai Karius[3], Florence Baudin [1], Helga Grötsch[1], Jan Kosinski [1,3] & Christoph W. Müller [1✉]

Transcription factor (TF) IIIC is a conserved eukaryotic six-subunit protein complex with dual function. It serves as a general TF for most RNA polymerase (Pol) III genes by recruiting TFIIIB, but it is also involved in chromatin organization and regulation of Pol II genes through interaction with CTCF and condensin II. Here, we report the structure of the *S. cerevisiae* TFIIIC subcomplex τA, which contains the most conserved subunits of TFIIIC and is responsible for recruitment of TFIIIB and transcription start site (TSS) selection at Pol III genes. We show that τA binding to its promoter is auto-inhibited by a disordered acidic tail of subunit τ95. We further provide a negative-stain reconstruction of τA bound to the TFIIIB subunits Brf1 and TBP. This shows that a ruler element in τA achieves positioning of TFIIIB upstream of the TSS, and suggests remodeling of the complex during assembly of TFIIIB by TFIIIC.

[1] European Molecular Biology Laboratory (EMBL), Structural and Computational Biology Unit, Meyerhofstrasse 1, 69117 Heidelberg, Germany. [2] Collaboration for joint PhD degree between EMBL and Heidelberg University, Faculty of Biosciences, 69120 Heidelberg, Germany. [3] Centre for Structural Systems Biology (CSSB), DESY and EMBL Hamburg, 22607 Hamburg, Germany. ✉email: christoph.mueller@embl.de

R NA polymerase III (Pol III) transcribes a number of abundant, short, folded RNAs in the eukaryotic cell, including tRNAs, 5 S rRNA, spliceosomal U6 RNA, and the signal recognition particle 7SL RNA. In yeast, transcription of all Pol III genes minimally requires two conserved transcription factors (TFs) to initiate transcription in vivo, namely TFIIIC and TFIIIB. *Saccharomyces cerevisiae* TFIIIC consists of six subunits with a combined molecular weight of 520 kDa, and binds to promoter elements termed A-box and B-box that are downstream of the transcription start site (TSS). The A-box element is located 12–20 nucleotides (nt) downstream of the TSS, and the B-box element is located 30–60 nt downstream of the A-box. TFIIIC then recruits TFIIIB and places it 25–30 nt upstream of the TSS (reviewed in ref. [1]). TFIIIB consists of subunits B-related factor 1 (Brf1), TATA-box-binding protein (TBP) and B double prime 1 (Bdp1), which together encircle the upstream promoter DNA in the pre-initiation complex and allosterically activate melting of the DNA double helix[2,3].

The six subunits of TFIIIC are organized in two subcomplexes: τA comprises subunit τ131, τ95, and τ55, and τB comprises subunits τ138, τ91, and τ60. τA binds the A-box with low affinity and is responsible for TFIIIB recruitment[4–6]. τB binds the B-box with high affinity, and the quality of the B-box determines promoter strength[6]. Due to the variable distances between A- and B-box elements found among tRNA genes, τA and τB are thought to be connected by a flexible linker[7–9].

In addition to its function as a Pol III TF, TFIIIC also plays important roles in shaping the three-dimensional (3D) organization of the genome, especially in higher eukaryotes. Human and yeast TFIIIC is recruited to *extra TFIIIC* (ETC) sites that vastly outnumber Pol III genes in humans[10–13]. ETC sites are enriched at the boundaries of topologically associating chromatin domains, at Pol II promoters, and close to the architectural protein CTCF[11,13]. TFIIIC has been demonstrated to recruit cohesin[14] and condensin II (ref. [15]), thereby regulating the formation of chromatin loops. Binding of TFIIIC to Alu-elements in the human genome has recently been demonstrated to control gene expression through chromatin looping and direct acetylation of histone tails[16].

Despite its importance as a general Pol III TF and architectural chromatin-organizing factor, the structure and molecular mechanism of holo-TFIIIC in Pol III recruitment have remained elusive, presumably due to its flexible nature. However, several crystal structures of TFIIIC subcomplexes and domains are available, including the N-terminal TPR array of τ131 (ref. [9]), a histidine phosphatase domain (HPD) of τ55 (ref. [17]), a heterodimeric portion of τ95–τ55 (ref. [18]), and a DNA-binding domain (DBD) of τ95 (ref. [18]). The structure of a τ60–τ91 heterodimer is also available[19], as well as a winged-helix (WH) domain of τ138 (ref. [9]).

Here, we have used a divide-and-conquer approach to further our structural understanding of TFIIIC. We report a 3.1 Å cryo-electron microscopy (cryo-EM) structure of τA, which constitutes the most conserved part of TFIIIC. We built an atomic model of τA that allowed us to locate mutations that affect TFIIIB binding and Pol III transcription. Our structural and biochemical studies identify an element of subunit τ95 that functions in auto-inhibiting DNA binding by τA. Lastly, we provide a negative stain map of τA bound to the TFIIIB subunits Brf1 and TBP—a complex that mimics a "probing" intermediate where the DNA is not yet engaged. We propose a model how a ruler element in τA positions Brf1 and TBP upstream of the TSS, where TFIIIB can assemble on a suitable DNA sequence by using a proofreading mechanism that contributes to TSS selection fidelity.

## Results

**Cryo-EM structure determination of τA.** We prepared recombinant τA and TFIIIC by co-expression of the respective subunits in insect cells. τA bound TFIIIB (prepared using the Brf1–TBP fusion protein[20] and wild-type (wt) Bdp1) during size-exclusion chromatography (SEC) and stimulated faithful in vitro transcription of a TFIIIC-dependent promoter, albeit at much lower levels than holo-TFIIIC (Supplementary Fig. 1). This resembles the in vitro transcription properties of tRNA genes lacking a B-box[6]. We collected cryo-EM data of a sample containing τA, TFIIIB, and a DNA fragment comprising the A-box and the TFIIIB-binding promoter elements of the His_tH(GUG)E2 gene. This allowed us to determine the structure of τA at an overall resolution of 3.1 Å (Supplementary Fig. 2). However, TFIIIB and DNA had dissociated from τA during sample preparation and are not visible in the EM density.

**Structure of τA.** The cryo-EM map allowed us to build an atomic model of τA with excellent refinement statistics (Supplementary Table 1), but also revealed density for two molecules of the detergent CHAPSO, which was added before plunge-freezing to prevent adsorption to the air–water interface[21], bound to τA (Supplementary Fig. 3). Our structure includes the N-terminal TPR array of τ131 (residues 131–573, the crystal structure of this construct has been described in ref. [9]) and a previously not described C-terminal domain (residues 612–1025), that contains a helical domain (residues 612–732) and an additional TPR array with seven repeats (residues 733–1025; Fig. 1 and Supplementary Fig. 4). The C-terminal domain packs against the N-terminal TPRs 6–10, resulting in an overall conformation of τ131 that resembles the letter "P" (Supplementary Fig. 4). τ131 acts as a scaffold for the assembly of τA, as described below.

The convex surface of the N-terminal τ131 TPR array binds the τ55 HPD and the triple β-barrel formed by the τ55–τ95 dimerization domains. Despite only being present in hemiascomycetes and being absent in other eukaryotes, the τ55 HPD is well-ordered and an integral part of the τA structure, rather than being flexibly attached to the τA core. We note that the active site of the τ55 HPD is solvent accessible in the context of τA (Fig. 1 and Supplementary Fig. 4).

τ95 is woven through the τA structure in an intricate manner. Its N-terminus originates between the τ55 HPD domain and the τ55–τ95 β-barrel. The β-barrel is followed by 75 amino acids (residues 161–236) that fold inside the superhelical groove of the C-terminal TPR of τ131 into a disc-like domain, containing two α-helices (Supplementary Fig. 5). This region connects the τ95 dimerization domain with the DBD. The τ95 DBD (residues 263–509) is positioned between the τ131 "ring" domain (residues 390–430, ref. [9]) and the C-terminal TPR array of τ131. Interestingly, a C-terminal *S. cerevisiae*-specific portion of τ95, which we refer to as the acidic plug (residues 566–592) is bound to the predicted DNA-binding interface[18] of the τ95 DBD (Fig. 2a and Supplementary Fig. 4).

**The C-terminus of τ95 auto-inhibits DNA binding of τA.** The acidic plug contains a helix that is rich in negatively charged residues and is embedded in the positively charged DBD. Because the DBD of the *Schizosaccharomyces pombe* τ95 homolog Sfc1 was shown to be auto-inhibited in DNA binding by a C-terminal acidic portion[18], we wondered if the *S. cerevisiae* C-terminal region, which comprises the acidic plug followed by an acidic disordered "tail", has a similar auto-inhibitory function in *S. cerevisiae* τA.

To test if the acidic plug or the acidic tail inhibit DNA binding, we prepared two τA variants with deletions in the C-terminus of τ95 (Fig. 2b). The first mutant, τ95Δ[plug], lacks the acidic plug and the disordered acidic tail (stop codon introduced after residue 521). The second mutant, τ95Δ[tail], lacks only the

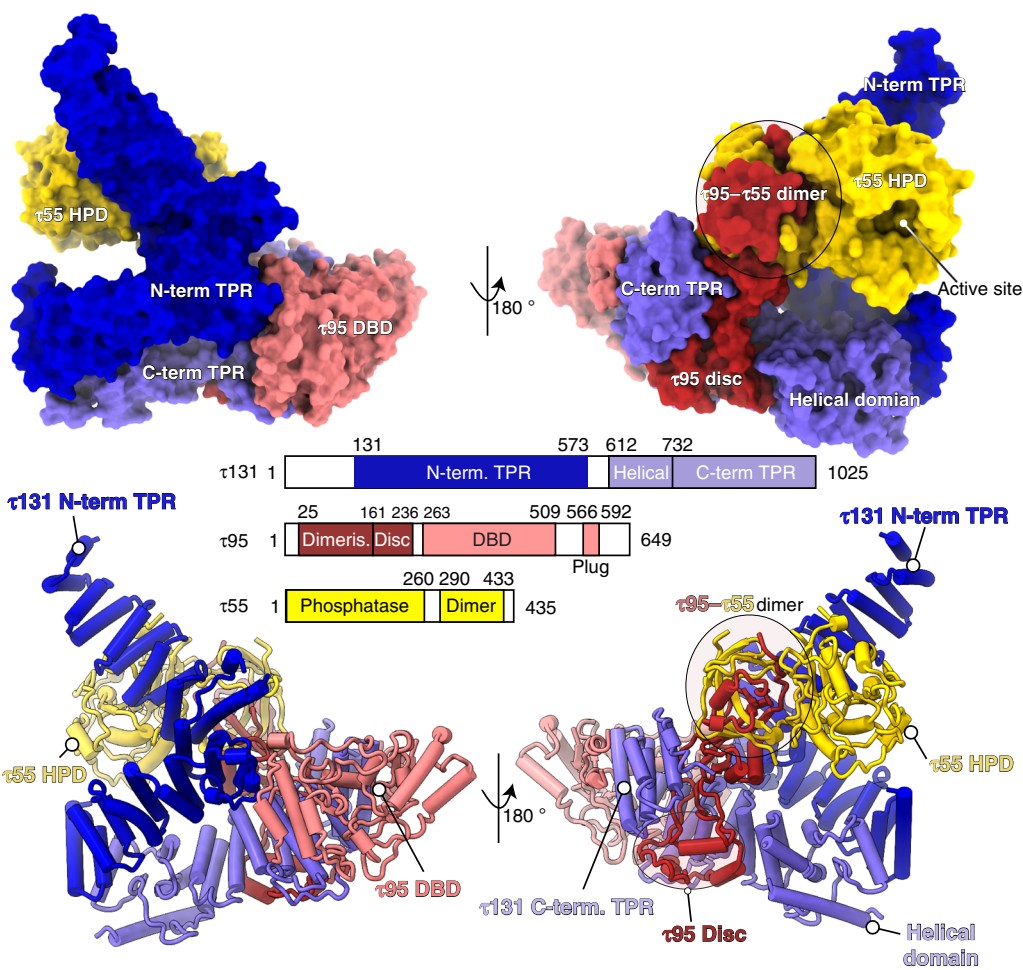

**Fig. 1 Structure of τA shown in two views, colored according to the bar diagram below.** The top panel shows τA as a surface rendering, and the bottom panel as a ribbon rendering with α-helices shown as cylinders.

disordered acidic tail (stop codon introduced after residue 592). Both variants bound A-box DNA in an electrophoretic mobility shift assay (EMSA) much stronger than full-length τA, confirming the auto-inhibition of DNA binding in the context of *S. cerevisiae* τA (Fig. 2c, left gel). However, the τ95Δ^tail mutant bound A-box DNA stronger than the τ95Δ^plug mutant. We obtained similar results using a filter-binding assay, but attempts to determine the affinity more precisely were hampered by the fact that we could not obtain τA at sufficient concentrations to achieve complete saturation of binding.

The data are consistent with a model where the acidic plug is not displaced from the DBD during DNA binding, but the acidic tail transiently associates with the positively charged DBD, thereby competing with DNA and reducing the affinity of τA to DNA. The relative reduction in DNA affinity of τ95Δ^plug compared to the τ95Δ^tail points to an architectural role of the plug in stabilizing τA rather than a role in DNA binding. Note that wt τA just begins to shift the probe at a protein concentration of 10 μM and a DNA concentration of 1 μM, suggesting that the affinity to the tested A-box sequence falls roughly in the micromolar range.

We next introduced the τ95Δ^tail mutation in recombinant holo-TFIIIC and tested DNA binding using EMSAs. Deletion of the acidic tail has only a mild effect on DNA binding by holo-TFIIIC under our experimental conditions (Fig. 2c). This indicates that in holo-TFIIIC, the effect of the tail mutation is buffered by the high-affinity interaction of τB with the B-box[6],

which presumably dominates initial formation of TFIIIC–DNA complexes and thus leads to a high local DNA concentration, which facilitates engagement of the A-box.

The Δ^tail constructs were also tested in in vitro transcription assays. Consistent with the results from our EMSA assays, deletion of the tail has only a minor stimulatory effect on the transcriptional activity of holo-TFIIIC in our experiments. Compared to wt τA, τAΔ^tail stimulated slightly higher transcription levels at increasing protein concentrations, but still has poor activity compared to holo-TFIIIC (Fig. 2d). This indicates that τB is required for full TFIIIC function, potentially due to the contribution of the τB subunit τ60 to TFIIIB recruitment[19].

We also introduced the τ95Δ^tail and τ95Δ^plug mutations in the endogenous TFIIIC locus in yeast to test their functional importance in vivo. We observe a growth defect for both mutations at elevated temperature on rich media (37 °C) and at optimal temperature on minimal media (30 °C; Fig. 2e). Therefore, the C-terminus of τ95 is functionally important in vivo. We speculate that the acidic tail might increase the specificity of the A-box interaction by outcompeting suboptimal DNA sequences, whereas the acidic plug appears to stabilize τA, consistent with its position at the interface of τ95 and τ131. Given that under exponential growth conditions, all tRNA genes are occupied by Pol III in yeast[10], perturbations that affect TFIIIC recruitment could easily lead to a reduction in the cellular tRNA pool, explaining the observed reduced growth rates of our τ95 mutant strains. This might not be captured in our in vitro transcription

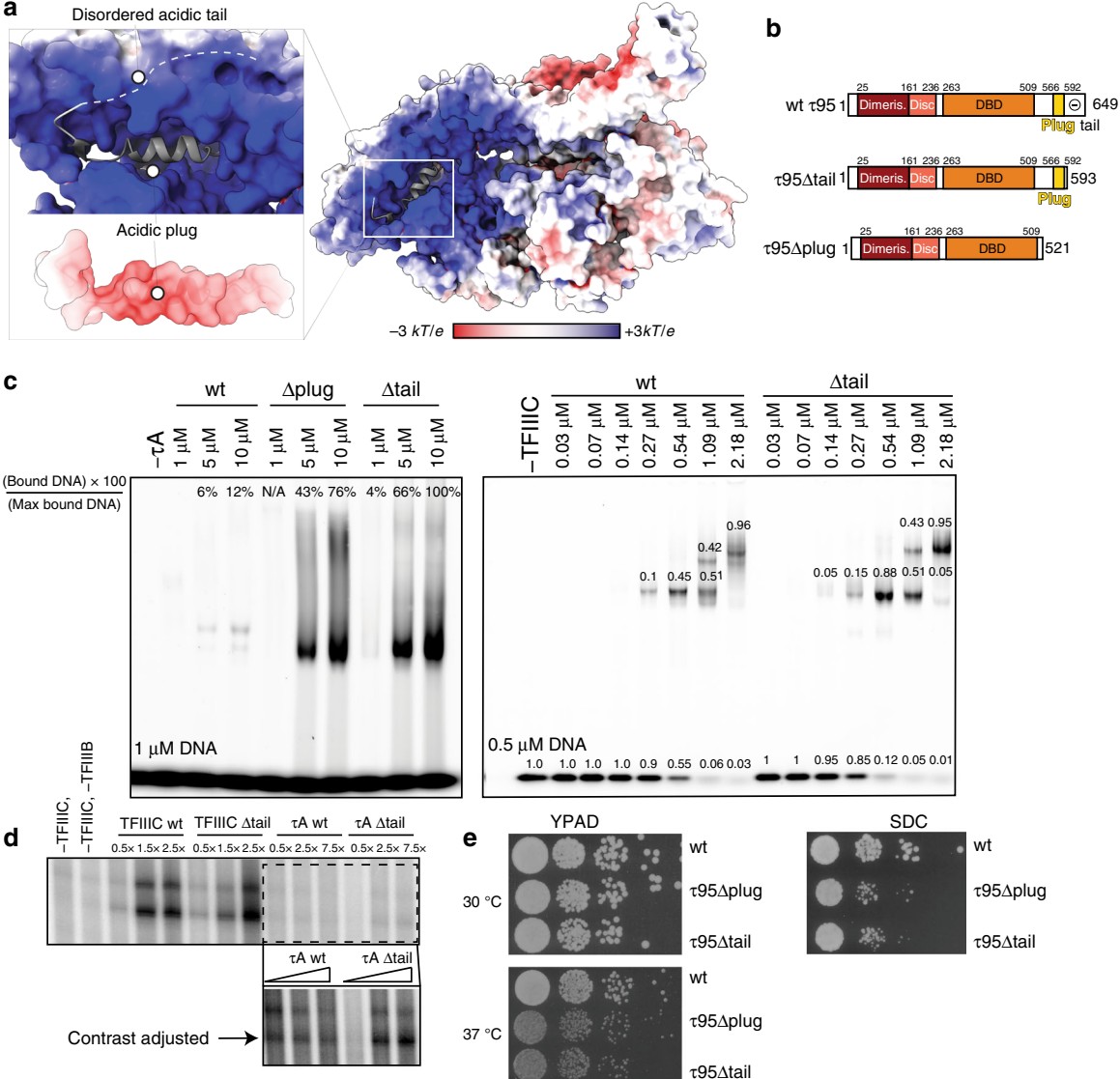

**Fig. 2 The acidic C-terminal region of τ95 auto-inhibits DNA binding of τA. a** Electrostatic potential mapped onto the surface of τA. The acidic plug is negatively charged and embedded in the positively charged DBD, followed by a disordered acidic tail (indicated by dashes). **b** Domain representation of τ95, τ95Δtail, and τ95Δplug. **c** EMSAs comparing the effect of deletions in the τ95-C-terminus in the context of τA (left) and TFIIIC (right). Left gel: labels (in %) at the top of lanes show the relative intensity of the bound DNA in that lane, normalized to the most intense band of bound DNA in the gel. Right gel: numbers above bands show the amount of DNA present in the band relative to the total DNA in the lane. **d** In vitro transcription assays comparing the effect of the Δtail deletion on transcriptional activity of τA and TFIIIC. The boxed region in the upper panel was cropped and the contrast was adjusted to better visualize differences between τA and τAΔtail. **e** Yeast viability assays of τ95Δplug and τ95Δtail strains compared to τ95 wt. Source data are provided as a Source data file.

assays due to the use of short DNA oligos as templates, and therefore lack of suboptimal, competing sequences.

**τ95 cannot bind DNA in a canonical way.** The τ95 DBD consists of a WH domain and a WH-interacting domain, as described for the *S. pombe* homolog Sfc1 (ref. [18]). We searched the Protein Data Bank (PDB) for the closest structural matches of the τ95 dimerization domain and the τ95 DBD using the DALI server[22], and found the Pol II TF TFIIFβ (τ95 DBD to the TFIIFβ WH domain with a Z-score of 5.4 and an r.m.s.d. of 1.8 Å, the τ95 dimerization domain to the TFIIFβ dimerization domain with a Z-score of 3.1 and an r.m.s.d. of 4.4 Å). This mirrors the similarity of *S. pombe* Sfc1 and TFIIFβ reported previously[18]. To test if τA might adopt a similar position in the

Pol III-PIC as TFIIF does in the Pol II-PIC, we superimposed τA onto the Pol II-PIC (PDB 5oqj[23]), using only the τ95 DBD and the TFIIFβ WH domain for alignment. While this illustrates the close structural similarity between the WH domains (Fig. 3a), it is clear that the τ95 DBD cannot bind to DNA as the TFIIF-WH domain does in the context of our τA structure. Both, the τ95 acidic plug and the τ131 C-terminal TPR array, severely clash with DNA in the complex modeled through superposition with TFIIF (Fig. 3b). TFIIF-like binding would thus require the τ95D DBD to dissociate from these elements, which together bury a large area (2265 and 2840 Å², respectively, calculated with COCOMAPS[24]), arguing against this scenario. The superposition of the τ95 WH domain with a representative set of 56 structures of WH domains in complex

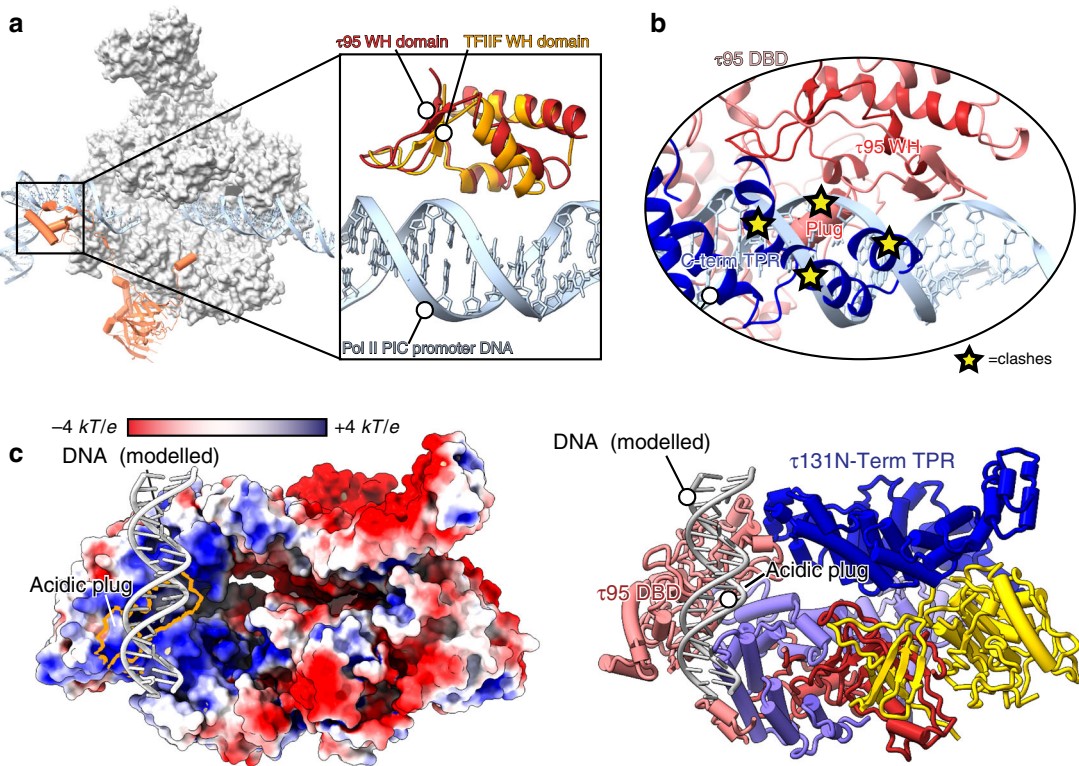

**Fig. 3 The τ95 WH cannot bind promoter DNA in the same way as the TFIIF-WH domain. a** Structure of the Pol II-PIC (PDB 5oqj), with TFIIFβ shown in orange and τ95 shown in red. For clarity, other transcription factors of the Pol II-PIC have been omitted. The boxed region shows a superposition of the τ95 WH and the TFIIF-WH domains (PDB 5oqj). **b** In the context of τA, the τ95 WH domain cannot bind DNA like the TFIIF-WH domain due to steric clashes with the acidic plug and the τ131 C-term TPR. **c** Model of possible DNA binding to τA, with τA shown as a surface colored by electrostatic potential (negative (red) to positive (blue)). The acidic plug (outlined in orange) does not interfere with the modeled DNA (left). A ribbon model is shown on the right with the same color code as in Fig. 1.

with DNA (retrieved from the CATH database[25] and curated from structures of Pol I, Pol II, and Pol III in complex with their TFs) also did not allow identifying a binding mode that could accommodate an unperturbed double-stranded DNA without steric clashes. Therefore, τA likely employs a binding mode that is different to the frequently observed WH domain–DNA interaction, where the recognition helix protrudes into the major groove. For example, it is conceivable that τA binds its promoter element in a shallow positively charged groove instead. To illustrate this, we placed B-DNA over the most positively charged surface of τA (Fig. 3c), but due to the possibility of conformational changes in τA during DNA binding and the lack of structural data this remains speculative.

**Recruitment of TFIIIB by τA.** The interaction between TFIIIC and TFIIIB has been extensively studied, leading to a sequential assembly model for TFIIIB (reviewed in refs. [1,26]). Assembly is initiated by recruitment of Brf1 to DNA-bound TFIIIC through the N-terminal TPR array of τ131. Next, Brf1 recruits TBP, and, finally, Bdp1 enters the complex, driven by binding sites in the τ131 N-terminal TPR array and in Brf1 and TBP. Bdp1 renders the TFIIIB–DNA complex extremely stable and resistant to high salt[27]. The recruitment process involves a series of conformational changes[28–31], and the initial step of the reaction, binding of Brf1 to the N-terminus of τ131, has been studied in detail. In particular, each of the two arms of the τ131 N-terminal TPR array is capable of binding Brf1[30]. In addition, gain-of-function mutations that increase Pol III transcription, loss-of-function mutations and mutations that rescue the loss-of-function mutations have been described in τ131 (refs. [29,32,33]).

In order to understand how TSS selection is achieved in the Pol III system, structural information about the complex of TFIIIB bound to τA/TFIIIC is necessary. However, as mentioned earlier, TFIIIB and DNA had dissociated from τA in our cryo-EM preparation. Interestingly, our cryo-EM map reveals a molecule of the detergent CHAPSO bound to the right arm of τ131, which is also predicted to bind TFIIIB (Supplementary Fig. 3), and it is therefore possible that addition of CHAPSO competed with TFIIIB for binding to τ131 and thereby contributed to dissociation of TFIIIB from τA. However, sample prepared without addition of CHAPSO aggregated strongly on EM grids and was unsuitable for data collection. Attempts to stabilize the sample through chemical crosslinking resulted in disruption of the complex, presumably because the crosslinker modified lysine residues in the DNA-binding interface. We thus prepared a complex of τA and the Brf1–TBP fusion protein and crosslinked the sample using the GRAFIX method[34]. Brf1–TBP bound τA with apparent 1:1 stoichiometry as assessed by co-elution on glycerol gradients in the absence of crosslinker (Supplementary Fig. 5). While attempts to obtain a high-resolution cryo-EM structure of the sample were unsuccessful, we obtained a negative stain reconstruction with a resolution of ~30 Å (Supplementary Fig. 6a). Note that our negative stain reconstruction is free of model bias, as we used the ab initio algorithm in CryoSPARC to obtain a 3D reference for particle alignment. We identified the most likely fit of the τA structure in this map, which led to the highest fitting scores with four different EM fitting metrics, and assigned the handedness of the map (see "Methods" section for details). Unfortunately, due to the low resolution, we could not unambiguously assign the orientation of Brf1 and TBP, although there overall location relative to τA can be extracted.

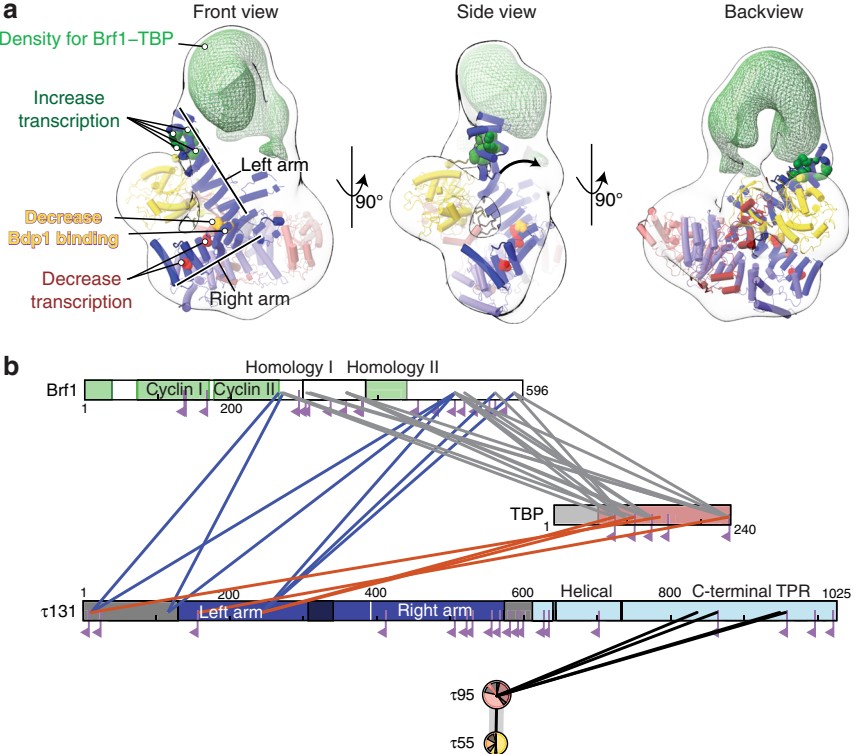

**Fig. 4 Interactions between τA and Brf1–TBP. a** Negative stain reconstruction of τA bound to Brf1–TBP. τA was fitted as a rigid body into the density. τA subunits are colored as in Fig. 1. Mutations in τ131 that affect Pol III transcription are shown as spheres and colored green (increased transcription), yellow (decrease Bdp1 binding) or red (decrease transcription). The arrow in the middle panel ("side view") indicates the movement of the τ131 TPR array that is required to fit into the density. Additional density for Brf1–TBP is shown as a green mesh. **b** Schematic view of high-confidence inter-subunits crosslinks of the τA–Brf1–TBP sample. Crosslinks between lysine residues with a score of >40 are displayed as solid lines. Mono-links are shown as flags. Domains of known structure in Brf1 are coloured green, parts of unknown structure/disorder are shown in white. τ131 domains are coloured as in Fig. 1. τ55 and τ95 domain diagrams are collapsed into circles.

Relative to the fit of τA, density for Brf1–TBP is located above the τ131 N-terminal TPR array, in agreement with previous reports that the N-terminal TPR is the main interaction hub of τ131 and Brf1 (refs. [26,29–31,35,36]). We observe that the τ131 N-terminal TPR array appears to be shifted toward this density (Fig. 4a, middle panel and Supplementary Movie 1). To assess if the observed shift is an artifact of negative staining, we also determined a negative stain structure of τA in absence of Brf1–TBP. The negative stain envelope of the "τA-only" map fits better to our cryo-EM structure (Supplementary Fig. 5e), supporting the idea that the N-terminal TPR is shifted in the Brf1–TBP-bound state relative to the apo state. Furthermore, this portion of τ131 also has the highest B-factors and lowest local resolution in the apo structure (Supplementary Fig. 4), and mutations that increase Pol III transcription map to this region[9,29] (Fig. 4a). Presumably, these mutations shift the equilibrium between the conformations observed in the apo state and the Brf1–TBP-bound state toward the latter. Interestingly, the positive effect on Pol III transcription of one of these mutations (Hl90Y located in TPR2) has previously been interpreted as functioning by relieving an auto-inhibited state of τ131 (ref. [31]); this auto-inhibited state could correspond to the τA conformation observed in the apo structure.

In contrast to τA, fitting of Brf1–TBP into the negative stain density led to several alternative fits with similar scores, not allowing us to identify a unique unambiguous fit. Nevertheless, visual assessment of a putative, high-scoring fit indicates that the additional density on top of τA can encompass the entire Brf1–TBP complex (Supplementary Fig. 6c). Crosslinking-mass spectrometry (XL-MS) restraints generated from the same sample (Fig. 4b) also did not allow us to unambiguously determine the orientation of Brf1–TBP in the negative stain map, because of the low number of detected crosslinks between structured Brf1–TBP and τA residues. Several factors might contribute to the difficulties in obtaining unambiguous fits. First, the available structures might change conformation upon complex formation, as seen for the τ131 N-terminal TPR. As only structures of Brf1–TBP in DNA-bound form are available, the relative orientation of the domains might be different in absence of DNA. Second, the complex might be flexible to a degree (discussed below), resulting in averaging of different conformations in the map. Third, the negative stain map appears to enclose a larger volume than expected when considering the portions of τA and Brf1–TBP of known structure. This could be due to ordering of flexible parts in the complex (~20% of τA and Brf1–TBP residues are not resolved in the available structures), which cannot be modeled based on low-resolution EM data.

Nonetheless, the overall location of the Brf1–TBP density relative to τA is informative even without explicitly fitting the orientation of Brf1–TBP. Density for Brf1–TBP is located above the N-terminal TPR array of τ131 and contacts mostly the first two TPRs (Fig. 4). Additional density extends toward the τ95 DBD. However, no density is present close to the "right arm" of the τ131 N-terminal TPR array, although this part (TPRs 6–10) has been shown to constitute a binding site for Brf1 with higher affinity than TPRs 1–5 (ref. [31]). The implications of this finding are discussed below.

## Discussion

Here, we report structural models for *S. cerevisiae* τA alone, and for τA bound to the TFIIIB subunits Brf1 and TBP in absence of DNA, which provide insights into yeast-specific and general features of τA. Both yeast and human TFIIIC have enzymatic activity, although with clear differences: the purified *S. cerevisiae* τ55 HPD domain has phosphatase activity and has been shown to be able to dephosphorylate peptides derived from the Pol II C-terminal domain that are phosphorylated on tyrosine 1 in vitro[17], while several human TFIIIC subunits possess histone acetyl transferase activity[37,38]. However, because the τ55–τ95 hetero-dimer exists independently of TFIIIC in yeast[39], it was unclear if TFIIIC harbors enzymatic activity or not. Our structure shows that the τ55 active site is accessible in τA, arguing that TFIIIC can directly modify substrates and the activity might be coupled to its function as a TF.

Our biochemical studies also confirm the existence of a dis-ordered, auto-inhibitory C-terminal tail of subunit τ95, which contributes to the low affinity of τA to DNA. We have previously shown that a negatively charged tail at the C-terminus of τ95 is conserved from yeast to humans[18], and our in vivo experiments show deletion of this tail confers a temperature-sensitive growth defect in *S. cerevisiae*.

TFIIIC serves as an assembly factor for TFIIIB and our XL-MS data obtained from a sample containing τA, and the TFIIIB subunits Brf1 and TBP recapitulate published interactions between τ131 and Brf1. We observe several crosslinks from the disordered N-terminus of τ131 to Brf1 and to TBP, in agreement with the N-terminus being necessary for high-affinity interaction between τ131 and Brf1 (refs. [9,30,40]). We also observe crosslinks between τ131 and the C-terminal part of Brf1, in agreement with the finding that a peptide corresponding to the Brf1 homology domain II inhibits assembly of TFIIIC–TFIIIB complex[35]. How-ever, at the chosen threshold (see "Methods" section), no cross-links are found between the Brf1 cyclin domains or the homology domain II and τ131, perhaps because lysine residues are buried in the interface and not accessible or not within crosslinking dis-tance to lysine residues in τ131. Unfortunately, this, together with the low resolution of the negative stain EM map, precludes us from modeling the exact position of these domains relative to τA.

Nonetheless, the overall position of TFIIIB relative to τA, and the elements of τA that are contacted by TFIIIB in our structure can be extracted and are informative with regard to TFIIIC's primary function in Pol III transcription, which is to assemble TFIIIB. While TFIIIB is bound preferentially ~30 bp upstream of the TSS, previous work[41,42] has demonstrated that TFIIIB pla-cement is codirected by TFIIIC and by direct interaction of the TBP subunit with the upstream DNA sequence. TBP can select a suitable AT-rich sequence within a 20 bp (or 70 Å) window[42], indicating some flexibility in the TFIIIC–Brf1–TBP complex. We refer to the assembly in which Brf1 and TBP are bound to TFIIIC, but not yet to the upstream sequence, as the "probing state". It has been suggested that τ131 is responsible for the flexibility in the probing state[42].

Our negative stain reconstruction of τA–Brf1–TBP lacks DNA, and therefore resembles the probing state, and thus an early assembly intermediate. This snapshot explains some of the bio-chemical observations concerning TFIIIC and assembly of TFIIIB made over the past decades; however, our interpretations remain to be validated by high-resolution structures.

First, we observe that the N-terminal TPR array of τ131 might be shifted in the Brf1-bound state relative to the apo state (Fig. 4a). This movement might explain how mutations in the N-terminal TPR array can increase transcription by shifting the equilibrium toward the conformation observed in the Brf1-bound state, thereby overcoming a rate-limiting step in the assembly of TFIIIB.

Second, we note that the negative stain density encloses a larger volume than expected when we only consider the regions of τA and Brf1–TBP that are ordered in available structures. It is hence possible that the flexible N-terminus of τ131, which is required for high-affinity binding to Brf1 (refs. [9,30,40]), becomes structured in the complex with Brf1–TBP. In line with this, it has been previously demonstrated through circular dichroism spectroscopy that τ131(residues 1–580), which encompasses the disordered N-terminus and the N-terminal TPR array, can form additional α-helices under certain conditions, and that binding to Brf1 to this construct also involves formation of new α-helices[30].

We would like to propose a model how additional binding sites in the τ131 TPR array, which are not contacted by TFIIIB in the probing state captured here, are used during later stages of TFIIIB assembly (Fig. 5). To our surprise, in our negative stain structure, Brf1–TBP is only bound to the left arm (TPR 1–5) of the τ131 TPR (Figs. 4 and 5a), but far away from the right arm (TPRs 6–10) of τ131, although the right arm constitutes a second binding site for Brf1 that has higher affinity than the left arm[31], and mutations that decrease Pol III transcription and Bdp1 binding also cluster to the right arm[9,36,43] (Fig. 4a). We thus believe that binding of DNA to Brf1–TBP and addition of Bdp1 to the complex triggers a large conformational change (in agreement with evidence from DNA footprinting data[44,45]) that brings the Brf1–cyclins and Bdp1 close to the right arm of τ131 (Fig. 5b). This might be initiated by initial bending of the DNA by TBP and subsequently stabilized through addition of Bdp1 (Fig. 5c and d). This two-step mechanism of TFIIIB assembly would open an opportunity to probing for an AT-rich upstream DNA element and allow for a proofreading mechanism, in which the lifetime of the initial TBP–DNA complex helps to select the correct sequence around which TFIIIB assembles. Consistent with this model, recent single-molecule experiments showed that the lifetime of the bent state of human TBP–DNA complexes depends on the quality of the TATA box, with suboptimal sequences having a shorter lifetime[46]. Hence, only a suitable AT-rich sequence could remain in the bent state for a sufficient amount of time to allow binding of Bdp1, whereas suboptimal sequences would revert to the unbent/disengaged state quickly. This mechanism could help to ensure the correct placement of TFIIIB and faithful TSS selection.

Our structures also hint at the mechanism by which TFIIIB is placed at a relatively constant position upstream of the A-box: the DBD auf τ95 and the Brf1–τ131 TPR array are located at opposite ends of τA. Therefore, τA might serve as a molecular ruler that positions TFIIIB at a fixed distance upstream of the A-box, where it can scan for a suitable "bendable" region. While structurally unrelated, this resembles the Pol II factor TFIID, which also binds to promoter elements within the transcribed region and assem-bles TBP upstream of the TSS.

Lastly, we would like to discuss the role of TFIIIC in transcrip-tion initiation. During the course of this project, substantial effort was invested in reconstituting a stable complex between TFIIIC, TFIIIB, and Pol III on different tRNA promoter sequences, yet without success. While the similar molecular weights of TFIIIC and Pol III make it difficult to interpret co-elution of TFIIIC and Pol III on size-exclusion columns or glycerol gradients, using recombinant τA we can clearly show that τA elutes separately from the TFIIIB–Pol III-his promoter complex when purified over a glycerol gradient or size-exclusion column (Supplementary Fig. 7), despite τA binding TFIIIB when Pol III was omitted (Supplementary Fig. 1). This suggests that τA, but also TFIIIC, act as assembly factors for TFIIIB, but are not bona fide components of the pre-initiation complex. Several findings are in line with this: when assembly of the Pol III-PIC on its promoter was monitored using chemical and enzymatic footprinting experiments, A-box protection

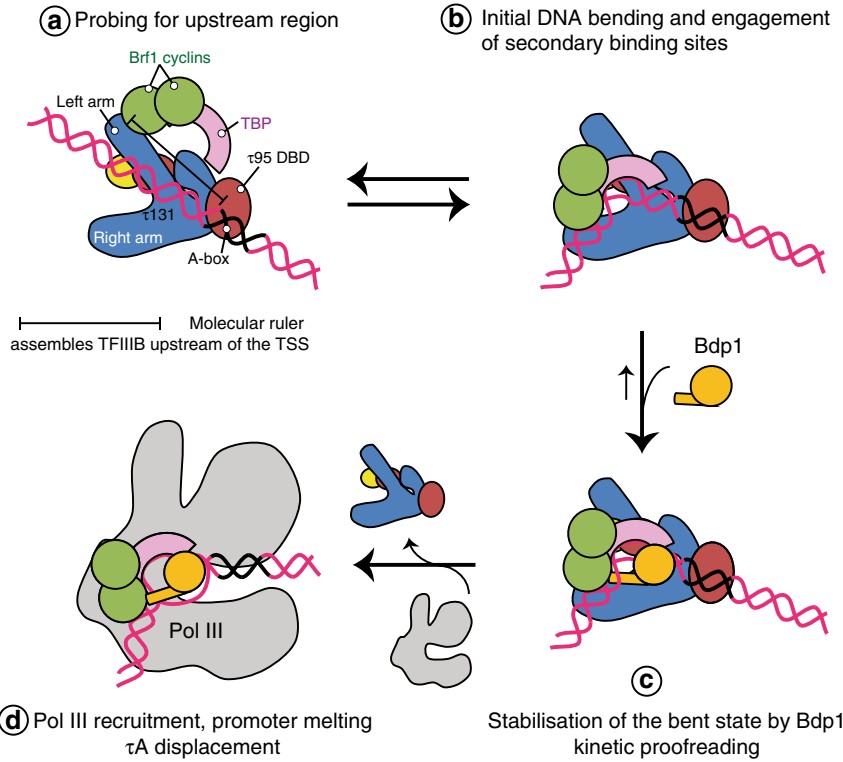

**Fig. 5 Model of TFIIIB assembly by τA/TFIIIC. a** Initially, Brf1 and TBP are recruited to the N-terminal TPR array of τ131. The distance between the τ95 DBD and the τ131 TPR array serves as a molecular ruler that places TFIIIB within a certain distance upstream of the TSS. **b** TBP binds and bends the upstream DNA sequence. The lifetime of this complex depends on the upstream DNA sequence engaged. **c** Bdp1 then enters the complex and stabilizes the bent state. **d** Recruitment of Pol III and promoter opening displaces τA, freeing the transcribed region.

was lost when Pol III was added[47], indicating that Pol III dissociates τA from its promoter element, although TFIIIC might still be tethered to the gene through binding of τB to the B-box. This agrees with previous in vitro transcription studies. Removal of TFIIIC did not affect transcription in vitro, once TFIIIB was assembled[48]. Chromatin immunoprecipitation studies have established that Pol III occupancy and TFIIIC occupancy are anticorrelated, with TFIIIC occupancy peaking under repressive conditions, and Pol III occupancy peaking under conditions that favor high transcription rates[11,12,49]. We speculate that displacement of τA/TFIIIC from its promoter concomitant with Pol III recruitment is a mechanism to ensure efficient promoter escape of Pol III. Interestingly, TFIID has recently also been proposed to be displaced from its promoter during transcription initiation[50], showing further parallels between the two factors.

## Methods

**τA expression and purification**. Insect cell codon-optimized genes for τA subunits were cloned into the pAceBac vector under a p10 promoter, and a TEV-cleavable his-tag was added to the N-terminus of τ95, using overlap extension PCR cloning. Expression cassettes were combined into the pBig1a plasmid using the BigBac[51] assembly method. Baculovirus was generated following standard procedures. Expression constructs for τ95ΔC mutants were generated via QuickChange mutagenesis PCRs.

τA and TFIIIC mutants were expressed in Hi5 insect cells. Cells were diluted 24 h before transfection at 5.10⁵ cells/mL. Infection was performed at a cell density of 1 million cells/mL and protein was expressed for 72 h at 27 °C. Cells were harvested at 800 × *g*, resuspended in PBS buffer and pelleted again. Cells were lysed in a 3× volume of lysis buffer (500 mM NaCl, 10% glycerol, 50 mM Tris-HCl pH 7.5, 4 mM β-mercaptoethanol (bME), 25 mM imidazole, 2 mM MgCl₂, 1:500 DNAse, 5 µL benzonase, and SigmaFast EDTA-free inhibitor tablets) by sonication. The lysate was cleared by ultracentrifugation for 1 h at 30,000 × *g* and filtered through 1.2 µm glasfiber filters, loaded on a pre-equilibrated 5 mL GE Healthcare Nickel NTA FF column. The column was washed with 75 mL of lysis buffer and eluted with His-B buffer (100 mM NaCl, 50 mM Tris-HCl pH 7.5, 4 mM bME, and 100 mM imidazole). Peak fractions were pooled, diluted to a conductivity of

~20 mS and loaded onto a 5 mL Heparin FF column pre-equilibrated in HepA buffer (100 mM NaCl, 50 mM Tris-HCl pH 7.5, 5% glycerol, and 5 mM DDT). Protein elution was performed with a linear gradient to 100% HepB buffer (1 M NaCl, 50 mM Tris-HCl pH 7.5, 5% glycerol, and 5 mM DDT). Peak fractions were diluted to 100 mM NaCl and loaded onto a MonoQ 5/50 GL column and eluted with a linear gradient from 0 to 100% HepB over 100 mL. This separated τA from excess τ95–τ55 dimer. The sample was concentrated and applied to a Superose 6 INCREASE 10/300 gel filtration column equilibrated in 150 mM NaCl, 20 mM HEPES pH 7.5, and 5 mM DTT.

**Expression and purification of TFIIIC**. For holo-TFIIIC, all six genes were cloned into a pBig2ab plasmid, with an N-terminal TEV-cleavable his-tag on τ95 and a C-terminal TEV-cleavable TwinStrep tag on τ138. Baculovirus was generated following standard procedures and TFIIIC was expressed in Hi5 cells as described for τA. TFIIIC was purified over Ni-NTA as described for τA, but eluted with 300 mM imidazol. The eluate was loaded on a 5 mL StrepTactin XP column (IBA Life Sciences), washed with 25 mL Strep-A buffer (150 mM NaCl, 50 mM Tris-HCl pH 7.5, and 5 mM DTT) and eluted with Strep-B buffer (Strep-A supplemented with 50 mM biotin). Peak fractions were concentrated and applied to a Superose 6 INCREASE 10/300 gel filtration column equilibrated in 150 mM NaCl, 20 mM HEPES pH 7.5, and 5 mM DTT.

**Expression and purification of Brf1–TBP and Bdp1**. The Brf1–TBP plasmid was transformed into BL21 Star (DE3) pRARE *E. coli* cells. Expression cultures were grown at 37 °C in TB medium to an optical density at 600 nm (OD) of ~1.0 and induced with 50 µM IPTG overnight at 16 °C. Cells were pelleted for 5 min at 12,000 g and resuspended in 3 mL lysis buffer (1 M NaCl, 50 mM Tris-HCl pH 7.5, 2 mM bME), 20% glycerol, 10 µg/mL DNase I, 1 x protease inhibitors (SigmaFast protease inhibitor cocktail EDTA free), 30 mM imidazole, 2 mM MgCl₂) per gram of pellet. After cell lysis, the lysate was cleared by centrifugation for 1 h at 30,000 g. Supernatant was incubated with 5 mL Ni-NTA resin (Qiagen) for 2 h. Beads were recovered and washed with 100 mL His-A buffer (1 M NaCl, 50 mM Tris-HCl pH 7.5, 2 mM bME, 5% glycerol, 30 mM imidazole) and 50 mL His-A low salt (His-A but with 150 mM NaCl) and eluted with 50 mL His-B (200 mM NaCl, 2 mM bME, 5% glycerol, 300 mM imidazole). The eluate was loaded on a 5 mL HiTrap Heparin column (GE healthcare) pre-equilibrated in HepA buffer (like His-B but without imidazole). The column was washed with 6 column volumes (CV) containing 30% HepB (HepA but 1 M NaCl) and eluted with a linear gradient from 30% HepB to 70% HepB over 20 CV. Brf1–TBP eluted at ~600 mM NaCl. Peak fractions were

concentrated and applied to a HiLoad 16/600 Superdex 200 size-exclusion column equilibrated in 300 mM NaCl, 25 mM HEPES pH 7.5, 5 mM DTT and 5 % glycerol. Purified Brf1–TBP was concentrated to ~6 mg/mL.

The Bdp1 plasmid30 was transformed in BL21 Star (DE3) pRARE *Escherichia coli* cells and grown in TB medium to an OD of ~1.0, cooled down, and induced with 100 μM IPTG overnight at 18 °C. Cell harvesting, lysis, and Ni-NTA chromatography were performed as for Brf1–TBP. Eluted proteins were loaded on a 5 mL Heparin column pre-equilibrated in HepA. The column was washed with 6 CV of 20% HepB and eluted with a gradient from 20% HepB to 70% HepB over 30 CV. Bdp1 (in ~520 mM NaCl) was cleaved with TEV protease overnight at 4 °C and incubated with 3 mL Ni-NTA for 1 h. Bdp1 was recovered using 100 mM imidazole buffer and purified by SEC as for Brf1–TBP (but in a buffer containing 150 mM NaCl).

**DNA oligonucleotides**. Experiments were performed with different lengths of the His_tH(GUG)E2 gene. For simplicity, only the sequence of the non-template strand is given here. All oligonucleotides were obtained from Sigma Aldrich. For all experiments, DNA was annealed by mixing equimolar amounts of the non-template and template strands in $H_2O$ followed by heating to 95 °C for 10 min before cooling to 20 °C at a rate of 1.5 °C/min.

For transcription assay, we used the sequence of the *His_tH(GUG)E2* gene, including 44 base pairs upstream of the TSS (referred to as *His_complete*, non-template 5′-GTATTACTCGAGCCCGTAATACAACAGTTCTCCATTGAAA AGTC**G**CCATCTTAGTATAGTGGTTAGTACACATCGTTGTGGCCGATGAA ACCCTGGTTCGATTCTAGGAGATGGCATTTT-3′, the +1 nt is printed in bold and A-box and B-box are underlined).

For TFIIIB–τA-binding experiments, we used *His_upstream_Abox* oligonucleotide (non-template strand 5′-AGCCCGTAATACAACAGTTCTCCATTGAAAAGT CGCCATC TTAGTATAGTGGTTAG-3′).

For TFIIIC–DNA-binding experiments, we used a *His_Abox_Bbox* oligonucleotide encompassing the A- and B-box elements. For the EMSA assay, the non-template strand had a 6FAM fluorophor attached to the 3′ end (non-template: 5′-CATCTTAGTATAGTGGTTAGTACACATCGTTGTGGCCGATGAAACCC TGGTTCGATTCTAGG6FAM-3′). For EMSAs with τA, we used an oligonucleotide encompassing only the A-box with a 6FAM fluorophor label on the 3′ end of the non-template strand (*His_Abox*, non-template: 5′-CATCTTAGTATAGTGGTTAGT6FAM-3′).

**Electrophoretic mobility shift assay**. τA EMSA samples contained 1 μM 6FAM labeled A-box DNA and 1 μM, 5 μM, or 10 μM τA construct in 100 mM NaCl, 20 mM HEPES pH 7.5, and 5 mM DTT.

TFIIIC EMSA samples contained 0.5 μM 6FAM labelled A-box–B-box DNA, and two-fold serial dilutions of TFIIIC, ranging from 2.18 μM to 0.03 μM.

Samples were incubated for 1 h on ice, supplemented with 10% glycerol and loaded on a 3–8% Tris-acetate gel (Thermo Fisher Scientific) in the running buffer: 2.5 mM Tris base, 19 mM glycine, and 1 mM DTT. Gels were imaged on a Typhon FLA9500 phosphorimager.

**τA–TFIIIB–Pol III reconstitution experiment**. A total of 108 pmol of His-DNA were incubated with an equimolar amount of τA for 15 min, followed by incubation with an equimolar amount of TFIIIB for another 15 min. The sample was diluted fourfold with SEC buffer (100 mM KCl, 20 mM HEPES pH 7.5, 5 mM DTT, and 2 mM $MgCl_2$) and an equimolar amount of Pol III was added. The sample is incubated overnight before being applied to a Superose 6 INCREASE 3.2/300 column equilibrated in SEC buffer or loaded on a 15–45% glycerol gradient. After centrifugation, glycerol gradients were manually fractionated by removing 200 μL fractions from the top of the gradient, and fractions were analyzed on 4–12% NuPAGE gels followed by silver staining.

**EM sample preparation**. τA was incubated with equimolar amounts of double-stranded *His_upstream_Abox* DNA, followed by the addition of the Brf1–TBP fusion protein and finally full-length Bdp1. The sample was incubated for 10 min before adding the next complex component, respectively. The complex was subsequently diluted to 1.3 mg/mL in dilution buffer (20 mM HEPES pH 7.5, 5 mM DDT, and 2 mM $MgCl_2$) and thereby adjusted to 75 mM NaCl. Directly before grid freezing 4 mM CHAPSO (3-([3-cholamidopropyl]dimethylammonio)-2-hydroxy-1-propanesulfonate in water) was added to the protein sample. A total of 2.5 μL of protein sample were applied to Quantifoil Cu 2/1 which had been previously glow discharged with a Pelco EasyGlow instrument. Excess liquid was removed with a Vitrobot Mark IV chamber (Thermo Fischer Scientific) at 4 °C, and 100% humidity for 4 s and at a blot force of 2.

**Cryo-EM and data processing**. Cryo-EM data were collected on a Titan Krios microscope with a Gatan Quantum energy filter and a K2 Summit direct detector in counting mode. For τA, data were collected at a magnification of 130,000× and a calibrated pixel size of 1.041 Å/px. We collected 5824 movies with an accumulated dose of 48.7 e−/Å² over 36 frames, using a target a defocus of −0.5 to −2 μm. Movies were preprocessed on the fly using wARP 1.06 (ref. [52]). The model parameters for motion correction and CTF estimation were set to 5 × 5 × 36 and

5 × 5 × 1, respectively. Particles were picked with BoxNet2_20180918 without retraining, using an expected diameter of 150 Å. Particles were inverted, normalized and exported in a 300 pixel box. Particles were divided into three batches, and each batch classified using 3D classification in RELION 3.0 (ref. [53]), using a 40 Å low-pass filtered negative stain model of τA as reference. The best class of each batch was retained, batches were joined and particles cleaned using 2D classification in RELION (setting the "Ignore CTFs until first peak" option to "yes"). Particles were refined, and then re-extracted from micrographs that were aligned using MotionCor2 and CTF corrected with gCTF to allow for Bayesian polishing inside RELION. Two rounds of CTF refinement (using per-particle defocus and beam tilt estimation) and Bayesian polishing were performed, yielding a map of 3.06 Å resolution.

**τA model building and refinement**. We initially placed the crystal structures of the τ131 N-term TPR (PDB 5aem) and the τ55 HPD (PDB 2yn0), as well as homology models of the τ55–τ95 β-barrel (modeled on PDB 4bjj) and the τ95 dimerization domain (modeled on PDB 4bji) into the density. Homology models were generated with Phyre2 (ref. [54]). A partial model of the C-terminal TPR array of τ131 was generated with the ARP/wARP[55] webserver. For this, we supplied the sequence of the C-terminal half of τ131 (residues 573–1025) and a partial map obtained by subtracting the fitted densities and removing noise, using the UCSF chimera[56] volume subtraction and volume eraser tools. ARP/wARP generated a backbone model of a large portion of the C-terminal TPR and also contained a stretch of correctly assigned sequence. Starting from this partial model, we manually completed the model in COOT[57].

Due to the high quality of the map and relatively small variations in local resolution, we could unambiguously assign the correct sequence to all density except of two connected α-helices that pack against residues 984–933 in τ131. We assign this density to subunit τ95, as it is located between residues 236–263, which are missing in our structure and have included the region as a poly-alanine model in the PDB file. The model was refined using ISOLDE[58] and phenix real-space refine[59].

**Negative stain of τA–Brf1–TBP**. For obtaining the negative stain map of τA–Brf1–TBP, we used a Brf1–TBP construct lacking the first 70 amino acids (corresponding to the Zn-ribbon and linker) because unlike Brf1–TBP, Brf1–TBPΔN did not form precipitates when diluted rapidly into low salt buffers. A total of 40 μg τA were mixed with equimolar amounts of Brf1–TBPΔN, and the sample was diluted to 75 mM NaCl and incubated for 15 min at 20 °C. The sample was then applied to a 10–30% (v/v) glycerol gradient containing glutaraldehyde. The gradient was prepared by layering 2.2 mL of heavy buffer (10% glycerol (v/v), 100 mM NaCl, 20 mM HEPES pH 7.5, 5 mM DTT, and 0.1% glutaraldehyde) underneath 2.2 mL of light buffer (10% glycerol (v/v), 100 mM NaCl, 20 mM HEPES pH 7.5, and 5 mM DTT) and rotating the tube for 55 s at an angle of 86° and a speed of 20 r.p.m. The gradient was allowed to settle for 2 h at 4 °C before the sample was applied on top of the gradient and centrifuged for 16 h at 150,000 × *g* in a SW55Ti rotor. The gradient was fractionated by removing 200 μL fractions from the top of the gradient by pipetting. Crosslinking was quenched by adding 10 μL 1 M Tris-HCl pH 7.5 to each fraction. Fractions were analyzed by SDS–PAGE and silver staining (when crosslinker was omitted) or InstantBlue staining (when crosslinking was performed). A total of 5 μL of sample from the indicated fraction in Supplementary Fig. 5 were applied to a freshly glow-discharged EM grid, the sample was incubated for 1 min on the grid, and then washed twice with water and then incubated for 1 min with 2% uranyl acetate solution. Excess uranyl acetate was blotted away with filter paper and the grid was allowed to dry. The sample was imaged on a Tecnai Spirit microscope operated at 120 kV at a nominal magnification of 49,000 and a pixel size of 2.292 Å. A total of 211 micrographs were collected with a target defocus of 1 μm. Micrographs were CTF corrected using GCTF and templates for autopicking in RELION were generated by an initial round of autopicking, using the Laplacian-of-Gaussian autopicker in RELION followed by 2D classification. A total of 10,731 template-picked particles were extracted from CTF-corrected micrographs with a 200 px box and cleaned through 2D classification in CryoSPARC. An initial model was generated from 7398 particles using the CryoSPARC ab initio reconstruction program. The model was then refined in RELION to an estimated resolution of 30 Å.

**Crosslinking-mass spectrometry**. A total of 40 μg of τA were incubated with an equimolar amount of the Brf1–TBP fusion protein at a protein concentration of 2.5 μM in 75 mM NaCl, 20 mM HEPES pH 7.5, and 5 mM DTT for 30 min at 20 °C. The sample was crosslinked for 1 h at 20 °C by addition of 5 mM H12/D12 isotope-coded di-succinimidyl-suberate (Creative Molecules) and quenched by addition 50 mM of ammonium bicarbonate for 10 min. Crosslinked proteins were denatured using urea and Rapigest (Waters) at a final concentration of 4 M and 0.05% (w/v), respectively. Samples were reduced using 10 mM DTT (30 min at 37 °C), and cysteines were carbamidomethylated with 15 mM chlorooacetamide for 30 min in the dark. Proteins were digested first using 1:100 (w/w) LysC (Wako Chemicals) for 4 h at 37 °C, and then the urea concentration was reduced to 1.5 M and digested was finalized with 1:50 (w/w) trypsin (Promega) overnight at 37 °C. Samples were acidified with 10% (v/v) TFA and desalted using OASIS® HLB

µElution Plate (Waters). Crosslinked peptides were enriched using SEC. In brief, desalted peptides were reconstituted with SEC buffer (30% (v/v) ACN in 0.1% (v/v) TFA) and fractionated using a Superdex Peptide PC 3.2/30 column (GE) on a 1200 Infinity HPLC system (Agilent) at a flow rate of 50 µL/min. Fractions eluting between 50–70 µL were evaporated to dryness and reconstituted in 30 µL 4% (v/v) ACN in 1% (v/v) FA. Collected fractions were analyzed by liquid chromatography-coupled tandem mass spectrometry (MS/MS). The mass spectrometric analysis was conducted using an UltiMate™ 3000 RSLCnano system (Thermo Fisher Scientific) directly coupled to an Orbitrap Fusion Lumos (Thermo Fisher Scientific). Peptides were loaded onto the trapping cartridge (µ-Precolumn C18 PepMap 100, 5 µm, 300 µm i.d. × 5 mm, 100 Å) for 5 min at 30 µL/min (0.05% TFA in water). Peptides were eluted and separated on an analytical column (nanoEase MZ HSS T3 column, 100 Å, 1.8 µm, 75 µm × 250 mm) with a constant flow of 0.3 µL/min using solvent A (0.1% formic acid in LC–MS grade water, Fisher Chemicals) and solvent B (0.1% formic acid in LC–MS grade acetonitrile, Fisher Chemicals). Total analysis time was 60 min with a gradient containing an 8–25% solvent B elution step for 39 min (min 6–45), followed by an increase to 40% solvent B for 5 min, 85% B for 4 min and 6 min of a re-equilibration step to initial conditions (2% B). The LC system was online coupled to the mass spectrometer using a Nanospray-Flex ion source (Thermo Fisher Scientific) and a Pico-Tip Emitter 360 µm OD × 20 µm ID; 10 µm tip (New Objective). The MS was operated in positive mode and a spray voltage of 2.4 kV was applied for ionization with an ion transfer tube temperature of 275 °C. Full scan MS spectra were acquired in profile mode for a mass range of 375–1600 $m/z$ at a resolution of 120,000 (RF Lens 30%, AGC target $2e^5$ ions, and maximum injection time of 250 ms). The instrument was operated in data-dependent mode for MS/MS acquisition. Peptide fragment spectra were acquired for charge states 3–7. Quadrupole isolation window was set to 0.8 $m/z$ and peptides were fragmented via CID (35% NCE). Fragment mass spectra were recorded in the ion trap at normal scan rate for a maximum of $2e^4$ ions (AGC target) or 100 ms maximum injection time. The instrument acquired MS/MS spectra for up to ten scans between MS scans. Dynamic exclusion was set to 60 s.

The data analysis was performed using xQuest and xProphet and all crosslinks at the 5% false discovery rate (FDR) were exported (Supplementary Data 1)[60]. To select an optimal crosslink score confidence threshold for analysis, the identified crosslinks were mapped to the τA and Brf1–TBP structures using Xlink Analyzer[61]. At the ld score (linear discriminant, as calculated by xQuest) threshold of 40, 25 crosslinks could be mapped to the structures with four of them exceeding the distance of 35 Å (two within τA and two within Brf1–TBP). Thus, only the crosslinks with the ld score at least 40 were selected for further analysis. We attribute the violated crosslinks to the flexibility of the complex in the absence of DNA and to the expected false positive identification rate at 5% FDR.

**Yeast tagging and viability assay.** To construct τ95Δplug and τ95Δtail yeast strains with a STOP codon after amino acid 521 or 592, respectively, C-terminal deletion cassettes with a ClonNat marker were amplified from pFA6a-NatNT2 (ref. [62]) and transformed into yeast BSY17 (ref. [63]). Colonies were selected on YPD plates containing 100 µg/mL of ClonNat, and analyzed by colony PCR and subsequent sequencing. For the viability assay, serial dilutions of the deletions strains or the parental strain were spotted on YPD medium.

**Fitting to the negative stain EM map.** To fit the τA and Brf1–TBP structures to the negative stain EM map and to determine the handedness of the map, we applied a procedure that explores a large number of possible fits and evaluates the fits, using four different EM fitting scores. First, we generated samples of alternative fits using the an unbiased global fitting approach based on FitMap tool from UCSF Chimera software[56], which places fitted structure at random locations in the EM map, optimizes the fits locally, clusters them by similarity, and keeps best scoring solution belonging to the cluster. Prior to the fitting, the structures were converted into a simulated EM map at the resolution of 33.5 Å to approximate the resolution of the negative stain EM map. The "overlap" score of Chimera (sum of aligned density products)[56] was used for fitting, as out of all scores available in Chimera this score is more suitable for negative density EM maps, as confirmed by its high correlation with the envelope scores used below. Because for EM maps the correct mirror image is unknown without additional information (such as a known reference map or atomic structure that fits unambiguously to a large portion of the map), both mirror maps were used for fitting. The fitting runs were performed using 100,000 random initial placements of the structures in the map and the requirement of at least 30% of the model map to be covered by the density envelope defined at the threshold of 0.06 with high clustering thresholds and the number of optimization steps reduced from the default 2000 to 100, which resulted in alternative 20,000–24,000 fits per structure per mirror map. The fits were then scored and ranked with alternative EM fitting scores: overlap, cross-correlation, Chamfer distance, and the envelope score, with the latter two selected due to their ability of scoring matches between two surfaces (an intensity threshold of 0.06 was selected for defining the envelopes of the negative stain map and the fitted map simulated from the query structure). The envelope scores correspond to the interpretation of a negative stain EM map as an approximation of the surface of the protein complex. The scores were calculated using UCSF Chimera[56] and TEMPy library[64]. The map shown in Fig. 4 (referred to as mirror map 1 in Supplementary Fig. 6) led to

higher scoring fits of τA for all four score types and resulted in the same top fit regardless of the score (Fig. 4 and Supplementary Fig. 6a), indicating the mirror1 as the correct image of the negative stain EM map. For neither of the mirror maps, however, the fits were statistically significant when evaluated as published previously[65]. Thus, for each mirror map, we generated pairwise combinations of the τA and Brf1–TBP fits, and assessed them with the above scores. For computational efficiency, to generate the pairwise combinations, the τA and Brf1–TBP structures were fitted again, but now with low clustering thresholds resulting in lower number of representative fits (25–50 fits per structure per mirror map after clustering). The combinations of these fits were then generated using Integrative Modeling Platform[66]. This led 1136 and 1200 combined fits, respectively, for mirror1 and mirror2, where a "combined fit" represents a candidate model, in which τA and Brf1–TBP are fitted to the EM map. Higher scoring combined fits were obtained again for the mirror map 1, giving additional confidence that we identified the correct handedness. The fitting yielded one unique fit of τA and several alternative fits of Brf1–TBP among the top scoring solutions (Supplementary Fig. 6b). The alternative fits of Brf1–TBP adopted the same location in the EM map relatively to the N-terminal arm of τ131, but different orientations (with some orientations rotated 180° to the others). Thus, as the representative fit of τA we selected the fit that (1) obtained the highest score out of all fits of τA, and for all four scoring measures, (2) agrees with the interactions from the τ131 arm to Brf1–TBP suggested previously (refs. [26,29–31,35,36]) and the crosslinking data from this work (Supplementary Fig. 6b). However, due to relatively low difference in EM fitting scores and because only three crosslinks between Brf1–TBP and τA could be mapped to the structurally resolved parts of the model (Supplementary Fig. 6b), the exact orientation of Brf1–TBP remained undefined, although its overall position in the map is defined by our confident identification of the τA fit (Supplementary Fig. 6c).

**Reporting summary.** Further information on research design is available in the Nature Research Reporting Summary linked to this article.

## Data availability

The cryo-EM map of the τA complex has been deposited to the Electron Microscopy Data Bank (EMDB) under the accession code EMD-10817. The coordinates of the corresponding model have been deposited to the PDB under accession code 6YJ6. The negative stain map has been deposited at the EMDB under accession code EMD-10795. The negative stain map as well as the fitted τA structure is also available as a chimera session (Supplementary Data 2). The MS proteomics data have been deposited to the ProteomeXchange Consortium via the PRIDE[67] partner repository with the dataset identifier PXD018232. Source data for Fig. 2 and Supplementary Figs. 1 and 7 are available with the paper online. Source data are provided with this paper.

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

## Acknowledgements
M.K.V. acknowledges support by the EMBL International PhD program. We thank H. Khatter for initial work on this project and W. Seifert-Davila for help with protein expression. We thank M. Rettel from the EMBL proteomics core facility for MS data acquisition and processing, and advice on XL-MS and deposition to the PRIDE database. We acknowledge W. Hagen and F. Weis for access to the EMBL Heidelberg cryo-EM platform, T. Hoffmann and J. Pecar for maintaining the high-performance computing environment at EMBL Heidelberg, and K. Remans for helpful discussions regarding insect cell expression construct design.

## Author contributions
C.W.M. initiated and supervised the project. M.K.V. designed and interpreted experiments, cloned and purified proteins, collected cryo-EM data, built the model, and prepared figures. A.J. cloned, expressed and purified proteins, and prepared EM samples. F.B. performed the Pol III transcription assays. H.G. helped with cloning, protein expression, and performed yeast tagging and viability assays. K.K. and J.K. performed fitting into the negative stain density and XL-MS analysis. M.K.V. and C.W.M. wrote the manuscript with input from the other authors.

## Funding

## Competing interests
The authors declare no competing interests.

## Additional information

**Peer review information** *Nature Communications* thanks Feixia Chu, Christoph Engel, and Kenji Murakami for their contributions to the peer review of this report. Peer review reports are available.

