## [Peer Review File · Nature Communications]

REVIEWER COMMENTS

Reviewer #1 (Remarks to the Author):

In this manuscript, the authors reported a 3.1 Å cryo-EM structure of τ A, the most conserved subcomplex of TFIIC that plays an important role in TFIIB recruitment and Pol III transcription. Though in the absence of DNA, structural information provides novel insights on the interactions of τ A with DNA, Brf1 and TBP subunits of the TFIIB complex. Further biochemical studies revealed a structural element of τ 95 that modulates interactions with DNA. Overall, the work is rigorous and the scope suitable for Nature Communications. Here are some suggestions and questions the authors might consider incorporating:

1. The τ A subcomplex cryo-EM structure was obtained with DNA and TFIIB dissociated during the preparation. Therefore, the structure might represent a state prior to DNA binding, maybe distinct from the DNA-bound state. Considering potential conformation changes, it might not be accurate to superimpose the structure of τ 95 to the DNA-bound Pol II-PIC structure (Fig. 3). Specifically, have the authors consider the possibility of a substantial conformational change in τ 95 during DNA binding, so that it adopts the canonical way to bind DNA?
2. It is not clear to this reviewer how the analysis of τ A-Brf1-TBP negative stain data, including the XL-MS data of the complex. When an unbiased ab initio modeling approach was used to reach conclusive assignment of the τ A, what was the orientation of Brf1-TBP in the top models? Was it completely random or was there a preference for the orientation of Brf1-TBP? Similarly, are XL-MS results consistent within τ A model? When mapped back to the top models, can XL-MS results provide discriminatory data for some Brf1-TBP orientations over others? Is there any reason why most detected crosslinkings are between Brf1 and TBP?
3. A minor typo in page 6, line 8, Extended data Fig "5" should be "4"; line 10, it should be "ring".
4. For Fig. 3, color code should be included in the figure legend. For Extended data Fig. 4, it will be nice to have the orientation of the subunits match to that of the τ A subcomplex in Fig. 1.

Reviewer #2 (Remarks to the Author):

In the present manuscript, Vorländer et al solve the structure of τ A, the subcomplex of TFIIC which is composed of τ 131, τ 95 and τ 55 and binds the A-box of the promotor and is responsible for TFIIB recruitment and transcription start site selection. τ 131 builds up the scaffold of the complex and contains two TPR arrays for protein-protein interactions, the active site of τ 55 HPD is also accessible within the complex and τ 95 is responsible for DNA binding with its DBD. A C-terminal acidic tail of τ 95, is shown to have an auto-inhibitory function in DNA-binding in vitro assays and analyzed in in vivo deletion experiments. Comparison of TFIIF and tau95 WH domain of indicates that DNA binding is apparently achieved by a different mechanism and a negative stain reconstruction of crosslinked τ A with TFIIB demonstrates that Brf1 is likely bound by the N-terminal TPR array of tau 131, explaining the distance and positioning of the promotor A-box and TATA-box.

This work by the Müller group is of high importance to the transcription community. The manuscript is well suited for publication in Nature Communications. I recommend publication with minor changes and would appreciate if the authors addressed one major question.

Major concern:

I wonder how exactly the sequence was assigned to the tau95 plug density? Could the authors sequence by cryo-EM? Generally, the 'plug'-density and primary sequence could be shown in Fig. 2 similar to Ext. Data Fig. 2C. Furthermore, complete TFIIC was only prepared and analyzed with the delta-tail version. Can the authors compare EMSAs and initial transcription also in the delta-plug background to underline their analyses and the suggestion, that "B-box interaction dominates initial

formation of TFIIC-DNA complexes"? How conserved is the plug sequence among organisms? Since the plug apparently stays attached to the DBD of tau95 during TFIIC promoter-engagement, I wonder if it has any influence in DNA-binding sequence specificity? Growth of delta-tail and -plug mutations could easily be tested under temperature or chemical stress conditions as the authors mention in the text. For clarity: Can the ratio of bound vs. unbound DNA be quantified in EMSAS? Is it clear why initial transcription of Pol III produces a product double-band in Fig. 2d?

Minor points.

Which domains of tauA are conserved among organisms? A simple supplemental Figure may be helpful to the general reader.

Can the plug be pointed out in Fig. 3c for clarity?

Page 6 line 8 (is the reference to extended data 5 or 4?)

What is exactly shown in Extended Data Fig. 2c? The section could be named and labeled.

The prominent presence of two CHAPSO molecules is shown in Ext. Data Fig. 3. Could this mimic any physiological interaction or lead to a bias of conformation by buffer components?

Simply out of interest: The authors use the position of TFIIF-beta WH to illustrate clashes of the tau95 WH with promoter DNA in Fig. 3. Can the position of the tandem-WH domain within the Pol I subunit A49 (on the other side of the upstream cleft as reported by the Müller group and others) be used as reference?

To me, it is not entirely clear what exactly the evidence for the proposed DNA binding mechanism of tauA is.

Adding the modelled DNA path and 2D labels for clarity in the supplemental movie may be helpful to readers (viewers).

Did the authors try to assemble their τ A-Brf1-TBP samples on a promoter scaffold to study later recruitment stages?

The Final results paragraph on interpretation of a tauA-TFIIB negative stain EM and XL/MS (pages 10 & 11) represents an important observation and is carefully and worded to reflect the speculative nature of assumptions. Some of these speculations may be better suited for the Discussion section.

Discussion line 2: Do the authors refer to sigma55 or tau55?

Methods: EM sample preparation: in buffer description "20 mM HEPES pH ?"

Reviewer #3 (Remarks to the Author):

This paper reports near-atomic resolution of tA structure, which is important in functions of pol III initiation. tA itself looks stable enough for Cryo-EM and XL-MS analyses, both of which are of high quality. Overall I see significant contribution to pol III transcription. However, some of their data and figures are not convincing to general readers who are not used to pol III transcription. For example, it is hard to follow their fitting structures into EM map of negatively stained sample (Fig. 4).

Transcription assays have poor signals (Fig. 2D). Hopefully some of following specific comments will help.

p.5 τ A bound TFIIB (prepared using the Brf1-TBP fusion protein²⁰ and wild-type (wt) Bdp1) stoichiometrically during size exclusion chromatography (SEC) and stimulated faithful in vitro transcription of a TFIIC-dependent promoter.

Relative amounts of tA to Brf1-TBP don't look same between fractions in the peak. I wonder if this is really stoichiometric complex or not (tA vs Brf1-TBP).

P5. However, TFIIB and DNA had dissociated from τ A during sample preparation and are not visible in the EM density.

Did it fall apart during grid preparation? I wonder if this assembly and if τ A prepared alone would yield the same EM density.

P5. but also revealed density for two molecules of the detergent CHAPSO, which was added before plunge-freezing, bound to τ A.

To prevent accumulation at air/water interface? If so, need reference.

P6. The acidic plug contains a helix that is rich in negatively charged residues.

The plug should be shown with side chains fitted into density.

P7. the disordered C-terminal tail is the auto-inhibiting portion of τ 95 rather than the plug.

This biochemical assay was carefully done but need to explain the discrepancy with previous results in pombe.

P7. We thus believe the acidic plug is not displaced from the DBD during DNA binding.

The dashed line for the tail in Fig. 2A is based on experimental data? Does the tail also contain negative residues? How does it inhibit DNA binding? Is there a role for the acidic plug at all? Does pombe also contain an acid tail?

P7. Note that wt τ A just begins to shift the probe at a protein concentration of 10 μ M and a DNA concentration of 1 μ M, suggesting that the affinity to the tested A-box sequence falls roughly in the mid-micromolar range.

I don't see any shift with WT τ A in Fig. 2C. I may see very faint band at this resolution. I think this statement is miswritten because wt τ A wasn't tested at [protein] = 10uM and [DNA] = 1uM. It was tested at [protein] = 10uM and [DNA] = 0.5uM.

P7. We speculate that the high-mobility species corresponds to TFIIC in which only the high-affinity B-box is engaged, whereas the low-mobility species represents TFIIC bound to the A- and B-box.

This explanation is not convincing or not necessary.

P7. We observe formation of the high-mobility species at approx. two-fold lower protein concentrations and a more homogenous shift to the low-mobility species in TFIIC t95Deltatail compared to wt-TFIIC.

This statement is overall confusing. There is formation of the high-mobility species at the same protein concentrations?

P7. The Δ tail constructs were also tested in in vitro transcription assays. Both TFIIC Δ tail and τ A Δ tail stimulated higher transcription levels at increasing protein concentrations, however τ A Δ tail was much less efficient than wt TFIIC (Fig. 2d).

This transcription assay is not convincing. I don't see enhanced signals with delta tail. First, is enhanced transcription expected with delta tail compared to wild type? What is the expected function of the tail? In other words, what is the function of the auto-inhibitory tail?

P8. We speculate that it might increase the specificity of the A-box interaction by allowing only binding of optimal A-box sequences, or it might be important to allow efficient promoter escape of Pol III by rapidly displacing τ A from the DNA, or have functions in regulating TFIIC activity in response to stress.

I see significant effect of the tail on the yeast growth rate at high temperature but it is hard to understand the following explanation about promoter escape, stress response etc.

P8. τ 95 cannot bind DNA in a canonical way

I didn't understand this paragraph well. Tau95 BDB and dimerization domain are supposed to be equivalent to those of TFIIF? I thought there are many WH domains in transcription and chromatin factors, which are variable in DNA-binding mode. Does TFIIF WH adopt canonical DNA binding? How does it compare to other WH domains in transcription?

P9. Brf1-TBP bound τ A with apparent 1:1 stoichiometry as assessed by co-elution on glycerol gradients in the absence of crosslinker (Extended Data Fig. 5).

In Fig. S5, the peak position looks different with and without fixation. If so, does the peak with fixation on the right have the same stoichiometry as that without fixation on the left?

P10. We identified the most likely fit of the τ A structure in this map and assigned the handedness of the map through the global fitting of the τ A structure and combinations of τ A and Brf1-TBP (see Methods).

I think their validation of fitting is hard to understand for general readers although I agree that the overall structure is consistent with XL-MS. Admitting that cryo-EM of this larger complex is not straightforward, why not proceed negative staining of τ A alone as a control? Using XL-MS of high quality in Fig. 4B, does integrative modeling work?

P10. Relative to the fit of τ A, density for Brf1-TBP is located above the τ 131 N-terminal TPR array, and we observe that the τ 131 N-terminal TPR array is shifted towards this density.

I am not sure if the observed structural change based on map from negatively stained sample is true.

P13. First, we observe that the N-terminal TPR array of τ 131 is shifted in the Brf1-bound state relative to the apo state (Fig. 4a).

I am not sure about this. This could be due to artifact of negative staining. Cryo-EM sample also included Brf1, although it was not observed.

P14. using recombinant τ A we can clearly show that τ A elutes separately from the TFIIB-Pol III-his promoter complex when purified over a glycerol gradient or size exclusion column (Extended Data Fig. 7).

Combined with Fig.S1, almost quantitative conversion in Fig. S7 is really impressive even to people who are not used to pol III biochemistry.

Point-by-Point Response

Reviewer #1 (Remarks to the Author):

In this manuscript, the authors reported a 3.1 Å cryo-EM structure of τ A, the most conserved subcomplex of TFIIC that plays an important role in TFIIB recruitment and Pol III transcription. Though in the absence of DNA, structural information provides novel insights on the interactions of τ A with DNA, Brf1 and TBP subunits of the TFIIB complex. Further biochemical studies revealed a structural element of τ 95 that modulates interactions with DNA. Overall, the work is rigorous and the scope suitable for Nature Communications. Here are some suggestions and questions the authors might consider incorporating:

1. The τ A subcomplex cryo-EM structure was obtained with DNA and TFIIB dissociated during the preparation. Therefore, the structure might represent a state prior to DNA binding, maybe distinct from the DNA-bound state. Considering potential conformation changes, it might not be accurate to superimpose the structure of τ 95 to the DNA-bound Pol II-PIC structure (Fig. 3). Specifically, have the authors consider the possibility of a substantial conformational change in τ 95 during DNA binding, so that it adopts the canonical way to bind DNA?

The reviewer is correct in pointing out that the conformation of τ A might change in response to DNA binding. However, the conformational changes required to bind DNA, for example, in the DNA-binding mode frequently observed for WH domains with the recognition helix protruding into the major groove would be enormous, due to the substantial clashes with τ 131, as shown in Fig 3c. Such binding would require the τ 95 DBD to dissociate from the rest of τ A. We have now included a buried surface area calculation to demonstrate the strong interaction of the τ 95 DBD with τ 131, making such a huge conformational change unlikely and arguing against a model in which the τ 95 DBD binds in this way. We also extended our analysis by inspecting 56 structures of WH domains bound to DNA, and we were unable to find a DNA binding mode that does not clash with the remainder of τ A. Finally, we now also emphasize that the τ A-DNA complex shown in Fig. 3c serves mostly illustrative purposes and remains speculative in the absence of structural data (see page 9).

“Both, the τ 95 acidic plug and the τ 131 C-terminal TPR array, severely clash with DNA in the complex modeled through superposition with TFIIF (Fig. 3b). TFIIF-like binding would thus require the τ 95 DBD to dissociate from these elements, which together bury a large area (2265 Å² and 2840 Å², respectively, calculated with cocomaps²⁵), arguing against this scenario. The superposition of the τ 95 WH domain with a representative set of 56 structures of WH domains in complex with DNA (retrieved from the CATH database²⁶ and curated from structures of Pol I, Pol II, Pol III in complex with their transcription factors) also did not allow identifying a binding mode that could accommodate an unperturbed double-stranded DNA without steric clashes. Therefore, τ A likely employs a binding mode that is different to the frequently observed WH domain-DNA interaction where the recognition helix protrudes into the major groove. For example, it is conceivable that τ A binds its promoter element in a shallow positively charged groove instead. To illustrate this, we placed B-DNA over the most positively charged surface of τ A (Fig. 3c), but due to the possibility of conformational changes in τ A during DNA binding and the lack of structural data this remains speculative.”

2. It is not clear to this reviewer how the analysis of τ A-Brf1-TBP negative stain data, including the XL-MS data of the complex. When an unbiased ab initio modeling approach was used to reach conclusive assignment of the τ A, what was the orientation of Brf1-TBP in the top models? Was it completely random or was there a preference for the orientation of Brf1-TBP?

We have extended the explanation of how the τ A-Brf1-TBP negative stain and XL-MS data were analyzed, also following the request of reviewer #3. For the orientation of Brf1-TBP in the top models, we had shown in the ED Fig. 6b and c that there is one orientation of Brf1-TBP that gives best fitting scores. Some of the alternative fits differ significantly from this fit (e.g. by approximately 180° rotations) but still occupy the same location in the map. To make it clearer in the revised version, we have made the alternative orientations better visible by enlarging the images of the structures and adding labels for Brf1 and TBP. We have also corrected one of the arrows in the ED Fig. 6b, which pointed to a wrong point in the plot. Despite that the top orientation leads to better scores, we still prefer to treat the top fit as tentative due to the relatively low difference in scores to the alternative fits.

Similarly, are XL-MS results consistent within τ A model? When mapped back to the top models, can XL-MS results provide discriminatory data for some Brf1-TBP orientations over others?

The XL-MS results are consistent with the model of τ A and our conclusion that Brf1-TBP binds to the N-terminal TPR array of τ 131. As stated in the methods, four crosslinks exceed the distance of 35 Å (two within τ A and two within Brf1-TBP), likely resulting from the flexibility of the complex. Unfortunately, at the chosen conservative crosslink score threshold, there are only three crosslinks between structured parts of τ A and Brf1-TBP. In the revised version, we mapped these three crosslinks to the models in the ED Fig. 6b. While they do seem to favor the orientation that also obtained the highest EM fitting scores, given the flexible nature of the complex, and the fact that two of the crosslinks stem from the same lysine residue in τ 131, this is in our opinion a too low number for unambiguously discriminating the orientation. Thus, as stated above, we treat the orientation of Brf1-TBP as tentative and limit our conclusions to localization of Brf1-TBP in the map but refrain from assigning an orientation within that volume.

Is there any reason why most detected crosslinkings are between Brf1 and TBP?

As exemplified by XL-MS data mapped to known structures (for example available in Xlink Analyzer database <https://www.embl-hamburg.de/XlinkAnalyzer/database.html>) only a minor fraction of possible crosslinks is usually observed. It is generally difficult to point out the specific reason even when the structure is known but the low number of crosslinks could be due to the absence of lysine residues within the appropriate distance and orientation, low reactivity of partially buried lysine residues, highly reactive lysine pairs “scavenging” the crosslinker from other residues, or unfortunate properties of crosslinked peptides (e.g. big molecular mass or high hydrophobicity) preventing detection by the applied chromatography and mass spectrometry protocols.

3. A minor typo in page 6, line 8, Extended data Fig “5” should be “4”; line 10, it should be “ring”.

The typing errors have been corrected.

4. For Fig. 3, color code should be included in the figure legend. For Extended data Fig. 4, it will be nice to have the orientation of the subunits match to that of τ A subcomplex in Fig. 1. In Fig.3 the color code has been included in the figure legend. The orientations in ED Fig. 4 were chosen to maximize visibility of the discussed domains and to illustrate the architecture of the individual subunits. We have therefore refrained from changing these depictions in ED Fig. 4, as some domains would overlap and make discerning them difficult when shown in the same orientation as in Fig. 1.

Reviewer #2 (Remarks to the Author):

In the present manuscript, Vorländer et al solve the structure of τA , the subcomplex of TFIIC which is composed of $\tau 131$, $\tau 95$ and $\tau 55$ and binds the A-box of the promoter and is responsible for TFIIB recruitment and transcription start site selection. $\tau 131$ builds up the scaffold of the complex and contains two TPR arrays for protein-protein interactions, the active site of $\tau 55$ HPD is also accessible within the complex and $\tau 95$ is responsible for DNA binding with its DBD. A C-terminal acidic tail of $\tau 95$, is shown to have an auto-inhibitory function in DNA-binding in vitro assays and analyzed in in vivo deletion experiments. Comparison of TFIIF and tau95 WH domain of indicates that DNA binding is apparently achieved by a different mechanism and a negative stain reconstruction of crosslinked τA with TFIIB demonstrates that Brf1 is likely bound by the N-terminal TPR array of tau 131, explaining the distance and positioning of the promoter A-box and TATA-box.

This work by the Müller group is of high importance to the transcription community. The manuscript is well suited for publication in Nature Communications. I recommend publication with minor changes and would appreciate if the authors addressed one major question.

Major concern:

I wonder how exactly the sequence was assigned to the tau95 plug density? Could the authors sequence by cryo-EM? Generally, the 'plug'-density and primary sequence could be shown in Fig. 2 similar to Ext. Data Fig. 2C.

As mentioned in the Methods section, the sequence register for the plug was obtained by automated model building using the Arp/Warp platform. Careful inspection of the result showed a good fit to the density and chemical environment, as well as no Ramachandran outliers, giving confidence that the sequence assignment is correct. The density fit is now displayed in ED Fig. 2c and each residue has been labeled. Note that acidic side-chains frequently lack density for their carboxylate moieties in cryo-EM maps due to radiation damage (Hattne et al, Structure 2018).

Furthermore, complete TFIIC was only prepared and analyzed with the delta-tail version. Can the authors compare EMSAs and initial transcription also in the delta-plug background to underline their analyses and the suggestion, that "B-box interaction dominates initial formation of TFIIC-DNA complexes"?

The rationale for testing only the $\Delta tail$ mutant in holo-TFIIC is that our analysis of the $\tau A \Delta tail$ and $\tau A \Delta plug$ mutants reveal no function of the plug in DNA binding (note that in the $\Delta plug$ construct, both the plug and tail elements are missing). The plug therefore appears to have a structural role in stabilizing the τA structure, in agreement with the reduced DNA affinity when comparing $\Delta tail$ and $\Delta plug$ mutants. Our statement that "B-box interaction dominates initial formation of TFIIC-DNA complexes" offers an explanation while introducing the $\Delta tail$ mutation in holo-TFIIC has a less drastic effect on DNA binding by TFIIC compared to τA . This is based on previously published observations that the B-box interaction (mediated by TFIIC subcomplex τB) has a much higher affinity than the A-box interaction (mediated by the TFIIC subcomplex τA), and is unrelated to comparing $\Delta tail$ and $\Delta plug$ mutants. Given that introducing the $\Delta tail$ mutation has a limited effect on DNA binding by holo-TFIIC, we expect that introducing the $\Delta plug$ mutation in holo-TFIIC will have an even less pronounced effect, given our EMSA data from the corresponding τA constructs. Therefore, the proposed experiments would in our opinion add very little to the manuscript.

How conserved is the plug sequence among organisms?

We have added a sequence alignment to ED Fig. 4. We do not observe conservation of the plug sequence. Therefore, the plug appears to be a structural, *S. cerevisiae*-specific element

that stabilizes the τ A fold. In line with this, the C-terminus of τ 95 is shorter in other species (see our new ED Fig. 4). We are now referring to this on page 6:

“Interestingly, a C-terminal S. cerevisiae-specific portion of τ 95, which we refer to as the acidic plug (residues 566-592) is bound to the predicted DNA-binding interface^s of the τ 95-DBD (Fig. 2a, Extended Data Fig. 4).”

Since the plug apparently stays attached to the DBD of tau95 during TFIIC promoter-engagement, I wonder if it has any influence in DNA-binding sequence specificity?

While we cannot exclude an effect of the plug on sequence specificity, we consider it more likely that the acidic tail influences sequence specificity by globally reducing affinity of τ A. We have now included a brief discussion about the potential functions of the plug and tail elements:

“Therefore, the C-terminus of τ 95 is functionally important in vivo. We speculate that the acidic tail might increase the specificity of the A-box interaction by outcompeting suboptimal sequences, whereas the acidic plug appears to stabilize τ A, consistent with its position at the interface of τ 95 and τ 131. Given that under exponential growth conditions all tRNA genes are occupied by Pol III in yeast¹⁰, perturbations that affect TFIIC recruitment could easily lead to a reduction in the cellular tRNA pool, explaining the observed reduced growth rates of our τ 95 mutant strains. This might not be captured in our in vitro transcription assays due to the use of short DNA oligos as templates, and therefore lack of suboptimal, competing sequences.”

Growth of delta-tail and -plug mutations could easily be tested under temperature or chemical stress conditions as the authors mention in the text.

We have now added the growth of Δ plug and Δ tail strains on minimal medium, where we also observe a growth defect, and we have updated Fig. 2 accordingly.

For clarity: Can the ratio of bound vs. unbound DNA be quantified in EMSAs?

We have added a quantification of bound vs unbound DNA for the holo-TFIIC EMSAs. Unfortunately, due to the low affinity of τ A, the τ A gels had to be exposed for a longer time to visualize the shifted bands, which led to a saturated signal for the unbound DNA. This precludes accurate quantification of the amount of unbound DNA for the τ A EMSA. Instead, we have compared band intensities for bound DNA for τ A samples, normalizing all values to the most intense band (10 μ M τ A Δ plug). The quantification is included in the updated Fig. 2.

Is it clear why initial transcription of Pol III produces a product double-band in Fig. 2d?

The appearance of a double-band appears to be a feature of the used promoter sequence and might originate from a cryptic transcription start site, which is often observed for *in vitro* transcription assays (compare also Sadian et al., Nat. Commun. 2019 for an example in Pol I transcription).

Minor points.

Which domains of tauA are conserved among organisms? A simple supplemental Figure may be helpful to the general reader.

The predicted domain architecture of the human counterparts of τ A subunits has been added to ED Fig. 4. Sequence identity between yeast and human domains is also indicated.

Can the plug be pointed out in Fig. 3c for clarity?

We have now labelled the plug in Fig. 3c.

Page 6 line 8 (is the reference to extended data 5 or 4?)

The reference is indeed to ED Fig. 4. The mistake has been corrected.

What is exactly shown in Extended Data Fig. 2c? The section could be named and labeled.

In our first submission, ED Fig 2c showed cryo-EM density for a helical element in τ 131 (residues 832-550), to illustrate the map quality. We have now replaced it with the EM density for the τ 95 plug, as requested by the reviewers.

The prominent presence of two CHAPSO molecules is shown in ED Fig. 3. Could this mimic any physiological interaction or lead to a bias of conformation by buffer components?

One of the CHAPSO molecules is located at the “right arm” of the τ 131 TPR array, and therefore overlaps with the expected binding site of TFIIB. It is therefore indeed possible that addition of CHAPSO might have contributed to dissociating TFIIB from τ A during sample preparation for EM. However, addition of CHAPSO was essential for obtaining a well-behaved sample for EM, as omitting it resulted in severe aggregation.

We have now added this observation on page 10:

“Interestingly, our cryo-EM map reveals a molecule of the detergent CHAPSO bound to the right arm of τ 131 which is also predicted to bind TFIIB (Extended Data Figure 3), and it is therefore possible that addition of CHAPSO competed with TFIIB for binding to τ 131 and thereby contributed to dissociation of TFIIB from τ A. However, sample prepared without addition of CHAPSO aggregated strongly on EM grids and was unsuitable for data collection.”

Simply out of interest: The authors use the position of TFIIF-beta WH to illustrate clashes of the tau95 WH with promoter DNA in Fig. 3. Can the position of the tandem-WH domain within the Pol I subunit A49 (on the other side of the upstream cleft as reported by the Müller group and others) be used as reference?

TFIIF beta was chosen in an unbiased approach, as it was identified as the closest structural match using the DALI server. We have now also used Pol I A49 as part of our more exhaustive analysis, but were still unable to find a DNA binding mode that wouldn't clash with τ 131 and the acidic plug. Please also see our response to Reviewer #1, point 1.

To me, it is not entirely clear what exactly the evidence for the proposed DNA binding mechanism of tauA is.

While the exact binding mode of τ A to DNA is unclear in absence of structural data, the depicted model is based on the electrostatic potential of the τ A structure as well as the identification of the τ 95 DNA-binding domain (DBD) in *S. pombe* in previous work. (Taylor et al, NAR 2013). We have now emphasized that the modelled DNA complex serves mostly illustrative purposes as stated on page 9:

“For example, it is conceivable that τ A binds its promoter element in a shallow positively charged groove instead. To illustrate this, we placed B-DNA over the most positively charged surface of τ A (Fig. 3c), but due to the possibility of conformational changes in τ A during DNA binding and the lack of structural data this remains speculative.”

Adding the modelled DNA path and 2D labels for clarity in the supplemental movie may be helpful to readers (viewers).

We have now added 2D labels to the movie. However, since the negative stain sample did not contain DNA, and we expect a large conformational change in presence of DNA, we consider the addition of modelled DNA to the movie could be misleading.

Did the authors try to assemble their τ A-Brf1-TBP samples on a promoter scaffold to study later recruitment stages?

We have compared complex stability of Brf1-TBP- τ A-hisDNA with that of TFIIB- τ A-hisDNA, and found that in the absence of Bdp1 the stability of the complex is significantly reduced. Since crosslinking was required to stabilize the complex for EM, but crosslinking also disrupted DNA binding, we have not further analyzed a Brf1-TBP- τ A-DNA complex.

The Final results paragraph on interpretation of a tauA-TFIIB negative stain EM and XL/MS (pages 10 & 11) represents an important observation and is carefully and worded to reflect the speculative nature of assumptions. Some of these speculations may be better suited for the Discussion section.

We have now moved this paragraph to the discussion section.

Discussion line 2: Do the authors refer to sigma55 or tau55?

We refer to τ 55. We thank the reviewer for spotting this mistake.

Methods: EM sample preparation: in buffer description "20 mM HEPES pH ?"

We thank the reviewer for spotting this mistake that has been corrected.

Reviewer #3 (Remarks to the Author):

This paper reports near-atomic resolution of tA structure, which is important in functions of pol III initiation. tA itself looks stable enough for Cryo-EM and XL-MS analyses, both of which are of high quality. Overall I see significant contribution to pol III transcription. However, some of their data and figures are not convincing to general readers who are not used to pol III transcription. For example, it is hard to follow their fitting structures into EM map of negatively stained sample (Fig. 4). Transcription assays have poor signals (Fig. 2D). Hopefully some of following specific comments will help.

p.5 τ A bound TFIIB (prepared using the Brf1-TBP fusion protein²⁰ and wild-type (wt) Bdp1) stoichiometrically during size exclusion chromatography (SEC) and stimulated faithful in vitro transcription of a TFIIC-dependent promoter.

Relative amounts of tA to Brf1-TBP don't look same between fractions in the peak. I wonder if this is really stoichiometric complex or not (tA vs Brf1-TBP).

The reviewer is correct in noticing that there are small shoulders to the peak, however the conclusion that the majority of TFIIB and τ A co-elute holds true. To address the observations, we have removed "stoichiometrically" from the sentence. Please note that Bdp1 and Brf1-TBP have virtually identical migration patterns in SDS-PAGE, leading to a higher intensity of the corresponding band compared to τ A subunits.

P5. However, TFIIB and DNA had dissociated from τ A during sample preparation and are not visible in the EM density.

Did it fall apart during grid preparation? I wonder if this assembly and if τ A prepared alone would yield the same EM density.

Given that TFIIB bound τ A during size exclusion chromatography, we assume that the complex fell apart during grid-preparation, as is often observed during sample preparation for cryo-EM.

P5. but also revealed density for two molecules of the detergent CHAPSO, which was added before plunge-freezing, bound to τA .

To prevent accumulation at air/water interface? If so, need reference.

The reviewer is correct. We have now added this additional explanation with a reference at page 5 of the revised manuscript.

P6. The acidic plug contains a helix that is rich in negatively charged residues.

The plug should be shown with side chains fitted into density.

The density fit of the acidic plug is now shown in ED Fig. 2c and the fitted residues have been labeled.

P7. the disordered C-terminal tail is the auto-inhibiting portion of $\tau 95$ rather than the plug.

This biochemical assay was carefully done but need to explain the discrepancy with previous results in pombe.

We are unaware of a discrepancy with the presented results to previous results in *S. pombe*. While previously a C-terminal portion of the $\tau 95$ -homolog Sfc1 in *S. pombe* was shown to auto-inhibit DNA binding in the isolated Sfc1 protein (Taylor et al, NAR 2013), our current study refines this model, showing that in *S. cerevisiae* the C-terminal region consists of a stably bound acidic plug, and a disordered tail. Comparison of wt, $\Delta(\text{plug+tail})$ and $\Delta(\text{tail})$ show that the tail, but not the plug, is responsible for inhibiting DNA-binding.

P7. We thus believe the acidic plug is not displaced from the DBD during DNA binding.

The dashed line for the tail in Fig. 2A is based on experimental data? Does the tail also contain negative residues?

The dashed line is not based on experimental data, but presented in order to help the reader. As indicated in Fig. 2a, b, the tail is also negatively charged. We have rephrased the corresponding section, which now reads:

*“... Because the DBD of the *S. pombe* $\tau 95$ homolog Sfc1 was shown to be auto-inhibited in DNA binding by a C-terminal acidic portion¹⁸, we wondered if the *S. cerevisiae* C-terminal region, which comprises the acidic plug followed by an acidic disordered ‘tail’, has a similar auto-inhibitory function in *S. cerevisiae* τA .*

To test if the acidic plug or the acidic tail inhibit DNA binding, we prepared two τA variants with deletions in the C-terminus of $\tau 95$ (Fig. 2b).”

How does it inhibit DNA binding?

Is there a role for the acidic plug at all?

We address the questions of the reviewer in a rephrased version of the paragraph (p.7) and on p.8. It now reads:

“However, the $\tau 95\Delta^{\text{tail}}$ mutant bound A-box DNA stronger than the $\tau 95\Delta^{\text{plug}}$ mutant. We obtained similar results using a filter binding assay, but attempts to determine the affinity more precisely were hampered by the fact that we could not obtain τA at sufficient concentrations to achieve complete saturation of binding.

The data is consistent with a model where the acidic plug is not displaced from the DBD during DNA binding, but the acidic tail transiently associates with the positively charged

DBD, thereby competing with DNA and reducing the affinity of τA to DNA. The relative reduction in DNA affinity of $\tau 95^{\Delta \text{plug}}$ compared to the $\tau 95^{\Delta \text{tail}}$ points to an architectural role of the plug in stabilizing τA rather than a role in DNA-binding”

...

Therefore, the C-terminus of $\tau 95$ is functionally important in vivo. We speculate that the acidic tail might increase the specificity of the A-box interaction by outcompeting suboptimal sequences, whereas the acidic plug appears to stabilize τA , consistent with its position at the interface of $\tau 95$ and $\tau 131$. Given that under exponential growth conditions all tRNA genes are occupied by Pol III in yeast¹⁰, perturbations that affect TFIIC recruitment could easily lead to a reduction in the cellular tRNA pool, explaining the observed reduced growth rates of our $\tau 95$ mutant strains. This might not be captured in our in vitro transcription assays due to the use of short DNA oligos as templates, and therefore lack of suboptimal, competing sequences.”

Does pombe also contain an acid tail?

As stated on p.6, Taylor et al (NAR 2013) identified a stretch of negatively charged residues in the *S. pombe* $\tau 95$ homolog Sfc1 that inhibits DNA binding by Sfc1. They have further shown that the negative charge of the $\tau 95$ C-terminus is conserved, but the primary sequence is not. While structural data of this portion of Sfc1 is not available, we consider it likely that the C-terminus of Sfc1 also forms a disordered tail. We now included a sequence alignment of the $\tau 95$ C-terminal region in ED Fig. 4e that includes *S. pombe*.

P7. Note that wt τA just begins to shift the probe at a protein concentration of 10 μM and a DNA concentration of 1 μM , suggesting that the affinity to the tested A-box sequence falls roughly in the mid-micromolar range.

I don't see any shift with WT τA in Fig. 2C. I may see very faint band at this resolution. I think this statement is miswritten because wt τA wasn't tested at [protein] = 10uM and [DNA] = 1uM. It was tested at [protein] = 10uM and [DNA] = 0.5uM.

The reviewer is correct in pointing out that the shifted band is very faint. We have now adjusted the contrast to better visualize the faintest bands. We have corrected the concentrations of DNA in Fig. 2c. EMSAs with τA were performed with [DNA]=1 μM (as stated in the Methods section).

P7. We speculate that the high-mobility species corresponds to TFIIC in which only the high-affinity B-box is engaged, whereas the low-mobility species represents TFIIC bound to the A- and B-box.

This explanation is not convincing or not necessary.
We have removed this statement

P7. We observe formation of the high-mobility species at approx. two-fold lower protein concentrations and a more homogenous shift to the low-mobility species in TFIIC $\tau 95^{\Delta \text{tail}}$ compared to wt-TFIIC.

This statement is overall confusing. There is formation of the high-mobility species at the same protein concentrations?

We have removed this statement and now emphasize that the deletion of the tail has a less dramatic effect on the affinity of TFIIC then it has on the affinity of τA . We have also added quantification of the band intensities to the corresponding Fig. 2c. The paragraph now states:

“Deletion of the acidic tail has only a mild effect on DNA binding by holo-TFIIC under our experimental conditions (Fig. 2c).”

P7. The $\Delta tail$ constructs were also tested in in vitro transcription assays. Both TFIIC $\Delta tail$ and $\tau A\Delta tail$ stimulated higher transcription levels at increasing protein concentrations, however $\tau A\Delta tail$ was much less efficient than wt TFIIC (Fig. 2d).

This transcription assay is not convincing. I don't see enhanced signals with delta tail. First, is enhanced transcription expected with delta tail compared to wild type?

Given the stronger binding of $\tau A \Delta tail$ in EMSAs, we speculated that $\tau A \Delta tail$ also supports higher levels of transcription. The analogous argument can be made for holo-TFIIC, although the difference in affinity, and consequently the expected difference in transcriptional activity, is smaller.

Following the suggestions of the reviewer, we have rephrased the corresponding section on page 7,8 to acknowledge better the limited effect of the $\tau 95 \Delta tail$ deletion on TFIIC. We also now emphasize that for full TFIIC activity, τB is required.

“The $\Delta tail$ constructs were also tested in in vitro transcription assays. Consistent with the results from our EMSA assays, deletion of the tail has only a minor stimulatory effect on the transcriptional activity of holo-TFIIC in our experiments. Compared to wt τA , $\tau A\Delta^{tail}$ stimulated slightly higher transcription levels at increasing protein concentrations but still has poor activity compared to holo-TFIIC (Fig. 2d). This indicates that τB is required for full TFIIC function, potentially due to the contribution of τB subunit $\tau 60$ to TFIIB recruitment¹⁹.”

What is the expected function of the tail? In other words, what is the function of the auto-inhibitory tail?

We have rephrased the section with our interpretation of these results. They now read on page 8:

“We observe a growth defect for both mutations at elevated temperature on rich media (37°C) and at optimal temperature on minimal media (30°C) (Fig. 2e). Therefore, the C-terminus of $\tau 95$ is functionally important in vivo. We speculate that the acidic tail might increase the specificity of the A-box interaction by outcompeting suboptimal DNA sequences, whereas the acidic plug appears to stabilize τA , consistent with its position at the interface of $\tau 95$ and $\tau 131$. Given that under exponential growth conditions all tRNA genes are occupied by Pol III in yeast¹⁰, perturbations that affect TFIIC recruitment could easily lead to a reduction in the cellular tRNA pool, explaining the observed reduced growth rates of our $\tau 95$ mutant strains. This might not be captured in our in vitro transcription assays due to the use of short DNA oligos as templates, and therefore lack of suboptimal, competing sequences.”

P8. We speculate that it might increase the specificity of the A-box interaction by allowing only binding of optimal A-box sequences, or it might be important to allow efficient promoter escape of Pol III by rapidly displacing τA from the DNA, or have functions in regulating TFIIC activity in response to stress.

I see significant effect of the tail on the yeast growth rate at high temperature but it is hard to understand the following explanation about promoter escape, stress response etc.

Please see our response to the previous point, where we have simplified our explanations.

P8. τ 95 cannot bind DNA in a canonical way

I didn't understand this paragraph well. Tau95 BDB and dimerization domain are supposed to be equivalent to those of TFIIF?

As shown previously for the *S. pombe* homolog Sfc1, and now in our manuscript for *S. cerevisiae* τ 95, the dimerization domain and the WH domain of TFIIF indeed closely resemble that of τ 95/Sfc1. Therefore, it is possible that the two proteins share a common ancestor. However, our structural modelling suggests that τ 95 cannot bind the DNA in a Pol III pre-initiation complex as TFIIF does in the Pol II pre-initiation complex.

I thought there are many WH domains in transcription and chromatin factors, which are variable in DNA-binding mode. Does TFIIF WH adopt canonical DNA binding? How does it compare to other WH domains in transcription?

Indeed, there are many WH domains that vary in their DNA-binding mode. Following the suggestion, we have checked whether any other DNA-binding mode could be accommodated without clashes by the τ A structure. To this end, we have superposed the τ 95 WH domain with a representative set of 56 structures of WH domains in complex with DNA (retrieved from the CATH database and curated from structures of Pol I, Pol II, Pol III in complex with their transcription factors. Despite different binding modes indeed present in this dataset, none of the binding modes involving unperturbed double-stranded DNA molecule could be accommodated without clashes with the acidic plug, τ 95 or τ 131. We have modified the text and the methods accordingly but kept the comparison to TFIIF, since it is the most similar DNA-bound WH domain we could identify.

P9. Brf1-TBP bound τ A with apparent 1:1 stoichiometry as assessed by co-elution on glycerol gradients in the absence of crosslinker (Extended Data Fig. 5).

In Fig. S5, the peak position looks different with and without fixation. If so, does the peak with fixation on the right have the same stoichiometry as that without fixation on the left?

The two gels with and without crosslinker in ED Fig. 5a do not show identical fractions in corresponding lanes (the experiments were done on different days with slightly different fractionation procedures). The gel without crosslinker has been included to demonstrate that Brf1-TBP binds τ A, whereas the gel on the right has been included to show that a peak fraction has been used to prepare the EM grids.

P10. We identified the most likely fit of the τ A structure in this map and assigned the handedness of the map through the global fitting of the τ A structure and combinations of τ A and Brf1-TBP (see Methods).

I think their validation of fitting is hard to understand for general readers although I agree that the overall structure is consistent with XL-MS.

We have added additional explanation and carefully revised the corresponding validation section in the Methods and the legend in ED Fig. 6 to make it more accessible to the general reader.

Admitting that cryo-EM of this larger complex is not straightforward, why not proceed negative staining of τ A alone as a control?

We thank the reviewer for the suggestion. We have now included a negative stain map of τ A alone and compare the fit of our cryo-EM structure to the apo τ A and the Brf1-TBP- τ A negative stain maps in ED Fig. 5.

Using XL-MS of high quality in Fig. 4B, does integrative modeling work?

As also explained in the response to the reviewer #1 (see above), unfortunately, there are too few crosslinks between τ A and Brf1-TBP for confident integrative modeling. At a conservative crosslink threshold, there are only three crosslinks between τ A and Brf1-TBP, and two of the crosslinks stem from the same lysine residue in τ 131. Accommodating the crosslinks through modeling would also require flexible fitting of the N-terminal arm of τ 131, which we prefer not to do given the low resolution of the negative stain EM map. Therefore we prefer to keep the analysis as it is without pinpointing the specific orientation but with the general conclusion that Brf1-TBP binds to the N-terminal arm of τ 131.

P10. Relative to the fit of τ A, density for Brf1-TBP is located above the τ 131 N-terminal TPR array, and we observe that the τ 131 N-terminal TPR array is shifted towards this density.

I am not sure if the observed structural change based on map from negatively stained sample is true.

We have toned down our statement (please see the next point).

P13. First, we observe that the N-terminal TPR array of τ 131 is shifted in the Brf1-bound state relative to the apo state (Fig. 4a).

We followed the suggestion of the reviewer and now compare the fit of our cryo-EM structure to the apo τ A and the Brf1-TBP- τ A negative stain maps in ED Fig. 5. We have also toned down our statement that the N-terminal TPR array is shifted.

“Relative to the fit of τ A, density for Brf1-TBP is located above the τ 131 N-terminal TPR array, in agreement with previous reports that the N-terminal TPR is the main interaction hub of τ 131 and Brf1 (Refs. ^{27,30–32,36,37}). We observe that the τ 131 N-terminal TPR array appears to be shifted towards this density (Fig. 4a, middle panel, Supplementary Video 1). To assess if the observed shift is an artifact of negative staining, we also determined a negative stain structure of τ A in absence of Brf1-TBP. The negative stain envelope of the ‘ τ A-only’ map fits better to our cryo-EM structure (Extended Data Fig 5e), supporting the idea that the N-term TPR is shifted in the Brf1-TBP bound state relative to the apo state. “

...

“First, we observe that the N-terminal TPR array of τ 131 might be shifted in the Brf1-bound state relative to the apo state (Fig. 4a).”

P14. “using recombinant τ A we can clearly show that τ A elutes separately from the TFIIB-Pol III-his promoter complex when purified over a glycerol gradient or size exclusion column (Extended Data Fig. 7).”

Combined with Fig. S1, almost quantitative conversion in Fig. S7 is really impressive even to people who are not used to pol III biochemistry.

We are very pleased that the referee finds these data convincing.

REVIEWERS' COMMENTS:

Reviewer #1 (Remarks to the Author):

The additional figures and discussions have addressed my previous comments and strengthened the manuscript.

Reviewer #2 (Remarks to the Author):

The manuscript by Vorländer et. al has improved during revision. Addition of most requested experiments, a new negative stain reconstruction of apo-tauA, comparative modelling, text and figure changes (especially Figs. 2; ED2 and ED4-6), correction of minor mistakes, and description of the 'plug' as specific to baker's yeast rounded up an interesting story. I have no further questions and recommend publication with Nature Communications.

Reviewer #3 (Remarks to the Author):

Their interpretations, in particular of biochemical data, are more appropriate and make it clearer to general readers without over simplification. Also their additional structure of negatively stained tA alone as a control (Fig. S5) looks good, and thus strongly supports the position of Brf1 and TBP on tA (although it seems they knew they are right.). Overall this is a decent solid paper, and should be accepted as it is.